# HIV-seq reveals gene expression differences between HIV-transcribing cells from viremic and suppressed people with HIV

Julie Frouard[1,2,10], Sushama Telwatte[3,4,10], Xiaoyu Luo[1,2], Natalie Gill [1], Reuben Thomas[1], Douglas Arneson[5], Pavitra Roychoudhury [6,7], Atul J. Butte [5], Joseph K. Wong [3], Rebecca Hoh[8], Steven G. Deeks [8], Sulggi A. Lee [9], Nadia R. Roan [1,2] ✉ & Steven A. Yukl [3] ✉

HIV-transcribing cells can perpetuate chronic inflammation in ART-suppressed people with HIV (PWH) and likely contribute to viral rebound after ART interruption. However, these cells are difficult to study using single-cell RNA-seq (scRNA-seq) due to their low frequency and low levels of HIV transcripts, which are usually not polyadenylated. By spiking in capture sequences targeting conserved regions of HIV during scRNA-seq – a new method we call "HIV-seq" - we detect double the mean number of HIV reads per cell from PWH. HIV RNA+ cells are enriched among T effector memory cells during both viremia and ART suppression but exhibit a cytotoxic signature during viremia only. In contrast, HIV-transcribing cells from ART-suppressed timepoints exhibit a distinct anti-inflammatory signature involving elevated TGF-β and diminished IFN signaling. These findings demonstrate that HIV-seq is a useful tool to better understand the mechanisms by which HIV-transcribing cells can persist during ART.

Most people with HIV (PWH) experience rebound of HIV in plasma within several weeks after stopping antiretroviral therapy (ART), indicating the persistence of a rebound-competent viral reservoir that prevents HIV cure. The prevailing model has been that the rebound virus arises from a small fraction of HIV-infected cells that contain an infectious provirus in a latent state, where the infected cell does not constitutively produce virions but can be induced to do so after activation[1–3]. However, the rebound virus is often different from that which grows out ex vivo following stimulation in quantitative viral outgrowth assays (QVOA)[4,5], suggesting that additional studies are needed to understand the reservoirs which can be reactivated in vivo.

A smaller body of research has focused on the cells actively transcribing HIV RNA in vivo, also known as the "active" reservoir. While HIV latency and expression are often viewed as a dichotomy (off/on), studies using multiple round QVOAs demonstrate varying degrees of inducibility ex vivo[6], and blood and tissue cells from ART-suppressed individuals show a continuum of HIV-expressing cells in vivo, with variable degrees of progression through different blocks to HIV expression[7–9].

Importantly, multiple lines of evidence suggest that HIV-infected cells which spontaneously express HIV RNA or protein in vivo may be just as important for pathogenesis and cure as the transcriptionally-

[1]Gladstone Institutes, San Francisco, CA, USA. [2]Department of Urology, University of California, San Francisco, CA, USA. [3]San Francisco Veterans Affairs (VA) Medical Center and University of California, San Francisco, CA, USA. [4]Department of Infectious Diseases, The University of Melbourne at the Peter Doherty Institute of Infection and Immunity, Melbourne, VIC, Australia. [5]Bakar Computational Health Sciences Institute, University of California, San Francisco, San Francisco, CA, USA. [6]Department of Laboratory Medicine and Pathology, University of Washington, Seattle, WA, USA. [7]Viral and Infectious Disease Division, Fred Hutchinson Cancer Research, Center, Seattle, WA, USA. [8]Division of HIV, Infectious Diseases and Global Medicine, University of California, San Francisco, CA, USA. [9]Zuckerberg San Francisco General Hospital and the University of California, San Francisco, CA, USA. [10]These authors contributed equally: Julie Frouard, Sushama Telwatte. ✉e-mail: nadia.roan@ucsf.edu; steven.yukl@ucsf.edu

silent reservoir. First, viral products expressed by active reservoir cells are likely to contribute to the immune activation and inflammation[10,11] that are thought to underlie the sequelae of ART-treated infection, including organ damage and reduced life expectancy[12–16]. Second, the active reservoir seems better poised than the silent reservoir to immediately infect new cells after interruption of ART because it does not need to revert the mechanisms that prevent viral expression in the silent reservoir. Indeed, at least four studies have shown that different forms of cell-associated HIV RNA negatively correlate with time to rebound after ART treatment interruption (ATI)[17–20]. Moreover, one small study showed that in about half of people who interrupted ART, *Pol* sequences from the rebound virus matched those from cell-associated HIV RNA prior to ART interruption[21]. These findings support the hypothesis that some part of the active reservoir is rebound-competent. Finally, active reservoir cells, by expressing HIV RNA and/or protein, are likely more susceptible than quiescent latent cells to new immune-based therapies aimed at an HIV cure (e.g., therapeutic vaccines, TLR agonizts, adoptive immune therapies, and broadly neutralizing antibodies).

Despite all this evidence pointing towards the active reservoir as an important target for HIV cure, our understanding of these cells is still rudimentary, and it is unclear whether these cells exhibit similar phenotypic features as HIV RNA+ cells present during viremia. Cells transcribing or translating HIV genes from viremic individuals have been characterized by flow cytometry, CyTOF, and single-cell sequencing[22–25] to a certain extent, but HIV-transcribing cells from ART-suppressed PWH have been harder to study with such single-cell technologies. For instance, conventional sequencing-based approaches identify active reservoir cells at such low numbers as to preclude meaningful analysis[25]. One alternative has been to activate cells from ART-suppressed individuals ex vivo to characterize reactivated HIV RNA+ cells[26,27], but this approach characterizes ex vivo-stimulated and not spontaneously HIV-transcribing cells. Of note, sequencing-based approaches for detection of HIV RNA+ cells from virally-suppressed PWH are not only limited by the low throughput and high costs of droplet-based encapsulation technologies (e.g. 10X Genomics), but also the reliance on RNA capture through poly-dT. Because most HIV-infected cells from ART-suppressed individuals do not contain poly-adenylated HIV RNA due to blocks to transcriptional elongation and completion[7], poly-dT-based methods of RNA capture will theoretically fail to recognize many HIV-transcribing cells.

To increase the ability to identify rare, HIV RNA+ cells from PWH, including in the context of ART suppression, we created a custom-modified, 10x Genomics-compatible, 5' sequencing-based scRNA-seq workflow in which the poly-dT primer is supplemented with multiple HIV-specific reverse primers targeting different regions of the genome. This approach increases the capture of HIV-infected cells with non-polyadenylated HIV transcripts, including 5' elongated as well as more distal HIV transcripts. In addition, we included DNA-barcoded antibodies (CITE-seq[28]) in our protocol to enable in-depth phenotyping of the HIV RNA+ cells for which we have transcriptome data. Compared to the standard 5' sequencing approach, the inclusion of HIV-specific reverse primers allowed the detection of more HIV transcripts from PWH, and enabled for the first time a meaningful analysis of HIV RNA+ cells from ART-suppressed PWH. Using this advantage, we performed an in-depth analysis of the transcriptomes and phenotypes of HIV RNA + cells from longitudinal samples of PWH during active viremia and after ART suppression.

## Results

### Development of HIV-seq as a method to increase capture and detection of HIV transcripts by single-cell sequencing

The standard 10X Genomics' 5' scRNA-seq workflow entails droplet encapsulation of individual cells, followed by capture and reverse transcription (RT) of polyadenylated transcripts using poly(dT) primers. This approach, theoretically, does not capture non-polyadenylated transcripts, although in practice some may be captured through non-specific binding. To more efficiently capture HIV transcripts, including those that are not poly-adenylated, we designed capture sequences targeting multiple conserved regions of the HIV genome (Fig. 1A): the R-U5-pre-*gag* region (for 5' elongated transcripts), the *pol* gene (for mid-transcribed, unspliced transcripts), the second exon of *tat-rev* (for distal transcripts, including spliced), and two regions known to be enriched among intact proviruses: the packaging signal[29] (Psi; for elongated, unspliced transcripts) and the HIV *rev* response element[29] (RRE, for distal and unspliced/single-spliced transcripts). Our capture sequences match the consensus sequence of subtype B HIV-1, and are known to be efficient reverse primers for established ddPCR assays[7]. The HIV capture primers were spiked into the poly-dT primer mix and used in the RT following cell encapsulation (Supplementary Fig. 2). After RT, gene expression and CITE-seq libraries were prepared and sequenced. We aligned all sequences to the GRCh38 human genome, to which we had appended a subtype B HIV-1 consensus sequence we had generated from the Los Alamos database (see Methods). We named our overall pipeline "HIV-seq" due to its specific targeting of HIV transcripts for RT, library generation, and alignment.

To assess the utility of HIV-seq, we compared it head-to-head to the original 10X Genomics' 5' single cell RNA-seq pipeline (without HIV primers). To obtain sufficient numbers of HIV RNA+ cells for this comparative analysis, we selected samples from two viremic donors (PID1052 and PID8027) not on ART and processed equal numbers of cells using the two methods. Similar numbers of total CD4 + T cells were captured and analyzed with the original pipeline vs. with HIV-seq (PID1052: 4263 and 8772 cells, respectively; PID8027: 8022 and 9701 cells, respectively). We first assessed the extent to which HIV-seq may perturb the capture and sequencing of the host transcriptome. UMAP visualization revealed similar gene expression profiles in the absence vs. presence of HIV-seq (Fig. 1B). Differential gene expression (DEG) analysis in the absence vs. presence of HIV-seq revealed 4 DEGs in one donor and 6 DEGs in the other, with only a single gene (long non-coding RNA AL138963.4) overlapping (Supplementary Fig. 3, Supplementary Tables 3 and 4), suggesting minimal perturbations to human gene expression profiling upon HIV primer addition.

Next, we determined whether HIV-seq can lead to spurious detection of HIV RNA transcripts in HIV-uninfected cells. We first determined whether HIV-seq led to the identification of any HIV transcripts in non-permissive CD8 + T cells, and found that it did not (Supplementary Fig. 4A). We then performed a side-by-side comparison of HIV-seq vs. conventional 5' scRNA-seq on PBMCs isolated from three HIV-uninfected donors. Similar numbers of total CD4 + T cells were captured and analyzed using the two methods (Donor 1: Conventional = 4668 cells, HIV-seq = 7260 cells; Donor 2: Conventional = 5417 cells, HIV-seq = 5998 cells; Donor 3: Conventional = 12,327 cells, HIV-seq = 8753 cells). Not a single HIV transcript was identified, irrespective of HIV capture primers (Supplementary Fig. 4B). Given the lack of background signal, this experiment also established that one HIV transcript could be used as the threshold for defining HIV RNA+ cells using HIV-seq.

Quantitative analyses showed that HIV-seq identified a higher percentage of HIV RNA+ cells from both donors than did traditional 5' scRNA-seq (Fig. 1C). For PID1052, HIV-seq increased the identification of HIV RNA+ cells from 0.047% (2 HIV RNA+ cells in 4263 CD4 + T cells) to 0.068% (6 HIV RNA+ cells in 8772 CD4 + T cells), corresponding to a 44.7% increase. For PID8027, HIV-seq increased the capture of HIV RNA+ cells from 0.77% (62 HIV RNA+ cells in 8022 CD4 + T cells) to 0.97% (93 HIV RNA+ cells in 9701 CD4 + T cells), corresponding to a 26% increase. While the small number of study

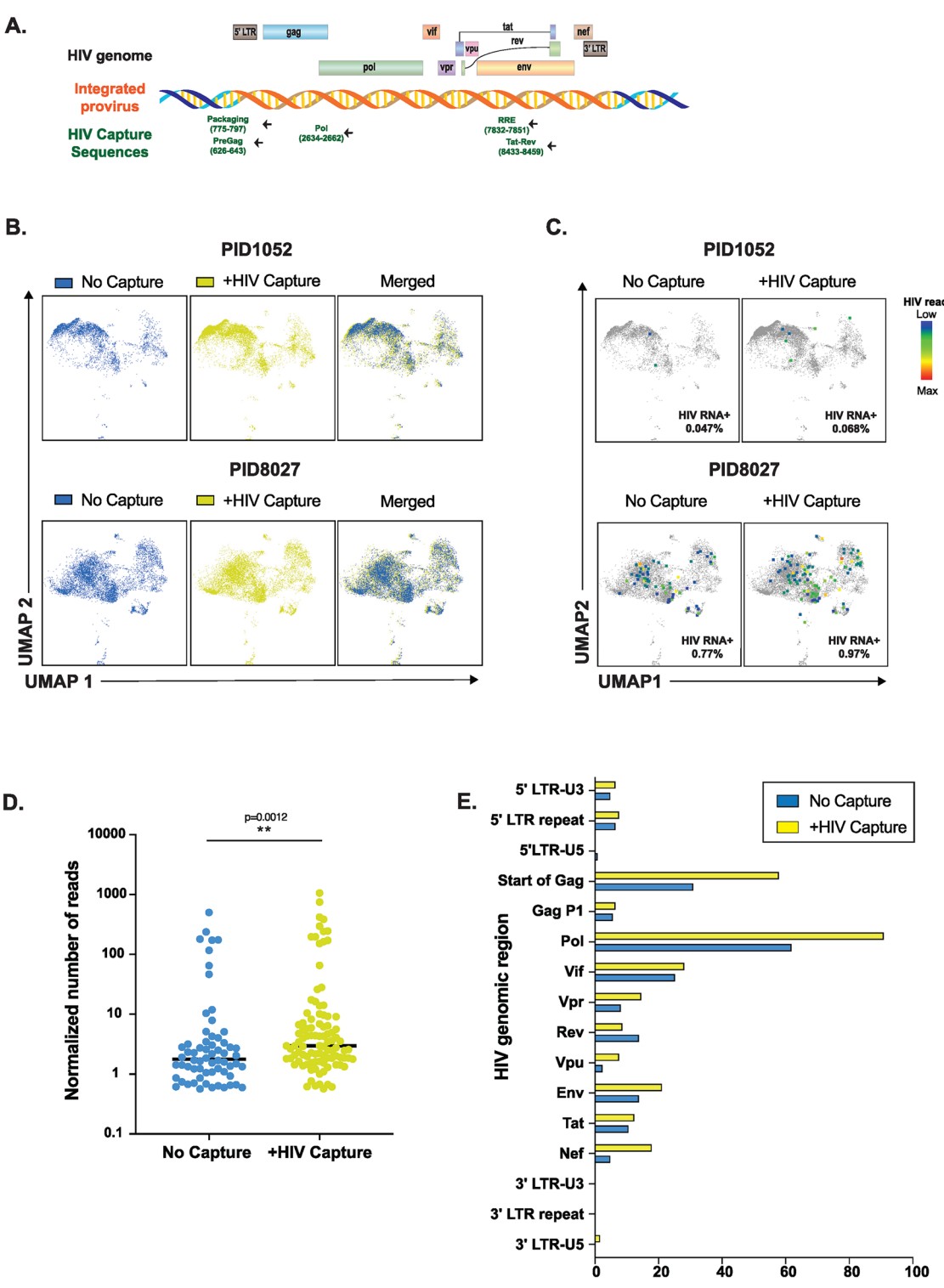

participants did not support formal statistical comparison of HIV RNA+ cell frequencies, HIV seq also detected a significantly higher number of HIV transcripts per infected cell (mean of 48 vs. 25 HIV reads/cell; P = 0.0012; Fig. 1D).

To assess whether the additional HIV transcripts captured by HIV-seq preferentially mapped to certain regions of the HIV genome, we compared the distribution of the HIV reads from the two viremic individuals in the absence vs. presence of HIV-capture primers. HIV-seq increased the detection of HIV transcripts from across the entire proviral genome, without changing the representation of the HIV regions detected relative to traditional 5′ scRNA-seq (Fig. 1E). Under both conditions, most HIV transcripts aligned to the *pol* and early *gag* regions of HIV. Application of HIV-seq in the context of ART suppression also resulted in detection of transcripts primarily mapping to the *gag* and *pol* regions (Supplementary Fig. 5). These results demonstrate that HIV-seq detects more HIV transcripts than conventional 5′ scRNA-seq and suggest that HIV-seq enables the detection of more HIV RNA+ cells.

**Fig. 1 | HIV-seq increases detection of HIV RNA+ cells from PWH without affecting host transcriptome. A** Schematic illustrating the genomic location and nucleotide position (based on HXB2) of HIV-specific capture primers for HIV-seq. **B** HIV-seq does not affect host transcriptome analysis by scRNA-seq. Shown are UMAP plots of CD4 + T cells from two ART-naïve viremic PWH (PID1052 and PID8027) processed in the absence (blue) vs. presence (yellow) of HIV capture sequences. **C** HIV-seq increases numbers of HIV RNA+ cells identified from viremic PWH. UMAP plots of scRNA-seq analysis of pre-ART CD4 + T cells from PID1052 and PID8027, showing HIV RNA+ cells (colored dots) among total CD4 + T cells (gray dots), with vs. without the addition of HIV capture sequences. Colors represent the number of HIV reads, from low (blue) to high (red).The percentages of HIV RNA+ cells among CD4 + T cells are indicated in the lower right of each plot. **D** HIV-seq

increases the numbers of HIV RNA reads detected per infected cell. Normalized numbers of HIV reads in the absence (blue) vs. presence (yellow) of HIV capture sequences are shown for each HIV RNA+ cell identified from PID1052 and PID8027 in panel B. Horizontal lines indicate medians. The analysis includes 64 cells detected without HIV capture oligos and 99 cells detected with HIV capture sequences. **P = 0.0012, as determined using a two-sided Mann-Whitney test. **E** HIV-seq increases detection of HIV reads throughout the HIV genome. HIV reads from PID1052 and PID8027 were aligned to the HIV-1 subtype B consensus reference genome. The y-axis depicts individual HIV regions, and the x-axis shows the normalized number of HIV transcripts per 10,000 CD4 + T cells, in the absence (blue) vs. presence (yellow) of the HIV capture sequences. Source data are provided as a Source Data file.

## HIV RNA+ cells from viremic PWH are preferentially cytotoxic Tem cells and exhibit an intracellular state promoting HIV replication

Leveraging the ability of HIV-seq to increase the numbers of HIV RNA+ transcripts and, to some extent, the number of HIV RNA+ cells we can analyze by scRNA-seq, we implemented it on cells from 4 viremic, ART-naïve PWH. We identified 8, 869, 155, and 40 HIV RNA+ cells out of 13,035, 35,849, 17,723, and 19,060 CD4 + T cells from PID1052, PID8026, PID8027, and PID145, respectively. In total, 1072 HIV RNA + CD4 + T cells were identified out of the 85,667 CD4 + T cells analyzed; infected cell frequencies ranged from 0.061% to 2.42% of the CD4 + T cell population (Fig. 2A). UMAP visualization of the transcriptomes of the HIV RNA+ cells revealed heterogeneity, in that infected cells were distributed in multiple regions of the UMAP space, yet enrichment was observed in some regions, suggesting non-random distribution of infected cells among CD4 + T cell subsets (Fig. 2B). Considerable variability in the degree of HIV transcription was observed among infected cells, with HIV transcript levels ranging from 1 to 1063 HIV reads per cell. When we separated the HIV RNA+ cells into those with low numbers of HIV reads (HIV$_{low}$: 1 to 50 HIV reads) and those with high numbers of HIV reads (HIV$_{high}$: >50 HIV reads), we found that participants with lower numbers of HIV RNA+ cells (PID1052 and PID0145) only harbored HIV$_{low}$ cells, while those with higher numbers (PID8026 and PID8027) had both populations (Supplementary Fig. 6A). Of note, HIV-seq did not affect whether HIV$_{high}$ could be detected (Supplementary Fig. 6B). The distribution of HIV$_{high}$ and HIV$_{low}$ cells across the UMAP was similar (Supplementary Fig. 6C) and the only significantly differentially expressed transcript/protein between these populations – apart from HIV RNAs – was the CD4 protein, which was decreased among the HIV$_{high}$ cells (Supplementary Fig. 6D). This finding likely reflects higher expression of Nef, which downregulates cell-surface CD4[30], in the HIV$_{high}$ cells. Because the HIV$_{high}$ and HIV$_{low}$ cells exhibited overall similar gene expression profiles, all remaining analyses combined these two populations together.

We then assessed whether the HIV RNA+ cells were enriched in specific cellular subsets. We first assessed the distribution of classical CD4 + T cell subsets (Supplementary Fig. 7) among uninfected and HIV RNA + CD4 + T cells. Relative to their uninfected CD4 + T cell counterparts, HIV RNA+ cells were under-represented among naïve cells and over-represented among memory cells (Fig. 2C), as expected[31]. Within memory T cells, HIV RNA+ cells were under-represented among central memory (Tcm) cells and those of the CCR7 + CD27- phenotype, and were over-represented among effector memory (Tem) cells (Fig. 2C), consistent with prior reports[25,32]. By contrast, transitional memory (Ttm), regulatory T cells (Treg), and T follicular helper (Tfh) cells were equally represented among uninfected and HIV RNA + CD4 + T cells (Fig. 2C).

Next, we compared differentially expressed proteins (DEPs) and transcripts between HIV RNA+ and HIV RNA- cells from the viremic PWH. At the protein level, HIV RNA+ cells expressed lower levels of the

naïve T cell marker CD45RA and higher levels of the memory T cell marker CD45RO compared to HIV RNA- cells across all donors (Supplementary Fig. 8), consistent with the notion that HIV RNA+ cells are enriched in memory CD4 + T cells (Fig. 1C). Other differentially expressed proteins were only observed in a subset of donors; these included CD279 (PD1; upregulated in HIV RNA+ cells in 2/4 donors), CD27 (downregulated in HIV RNA+ cells in 3/4 donors), and CD49d (upregulated in 3/4 donors) (Supplementary Fig. 8).

At the RNA level, almost 300 genes were differentially expressed between HIV RNA+ and HIV RNA- cells from the viremic PWH (Fig. 2D, Supplementary Table 5). Consistent with disenrichment of Tcm among HIV RNA+ cells (Fig. 2C), transcript levels of the CCR7, SELL (encoding the protein CD62L), and CD27 – markers of Tcm cells[33,34] – were downregulated in the HIV RNA+ cells (Fig. 2D). HIV RNA+ cells also had low transcript levels of the alarmins S100A8 and S100A9, which encode for S100 calcium binding protein A8 and A9, respectively. These proteins are released in response to environmental triggers and cellular damage[35]. Antiviral factors, including SERPINA1[36,37] and APOBEC3A[38], were also decreased among HIV RNA+ cells, suggesting that the intracellular state of HIV RNA+ cells in viremic individuals may favor HIV replication. In line with this finding, HIV RNA+ cells also exhibited downregulation of CST3, which encodes cystatin C, a cysteine protease inhibitor that interacts with the HIV proteins gp160, gp120, p31 and p24, and inhibits HIV protease function[39]. With regards to transcripts upregulated among HIV RNA+ cells, HIV transcripts were the top hit, as expected. In addition, HIV RNA+ cells expressed higher levels of the activation marker CD70. Interestingly, CD4 + CD70+ cells are increased in PWH with high levels of viremia and associate with immune activation[40], suggesting that this subset of infected cells may contribute to disease progression during untreated infection. HIV RNA+ cells also expressed elevated levels of CXCR6, which encodes a chemokine receptor that is an alternative co-receptor for HIV [41].

Pathway analysis of the DEGs (Fig. 2E) revealed upregulation of the NFAT pathway (MAF, CLTA4) in HIV RNA+ cells, consistent with NFAT as a driver of HIV transcription[42,43]. Similarly, there was an upregulation of the PKC pathway, known for its involvement in HIV gene expression and latency reversal[44]. Finally, and consistent with the DEG analysis, chemokine signaling pathways – featuring genes, such as CXCR6, the CXCR6 ligand CXCL13, and PLCB1 – were also elevated among HIV RNA+ cells.

While DEG analysis identified both known and novel shared features among all HIV RNA+ cells, it was clear that the HIV RNA+ cells were heterogeneous (Fig. 2B). We therefore implemented clustering to assess for transcriptomic or phenotypic features that may not be shared by the entire population of infected cells. Louvain clustering identified thirteen clusters of CD4 + T cells (Fig. 2F). The classical CD4 + T cell subsets (Supplementary Fig. 7) did not define the clusters (Supplementary Fig. 9A), and in fact all the classical memory subsets were represented among the 13 clusters, albeit in different proportions (Supplementary Fig. 9B). The 13 clusters could be differentiated by

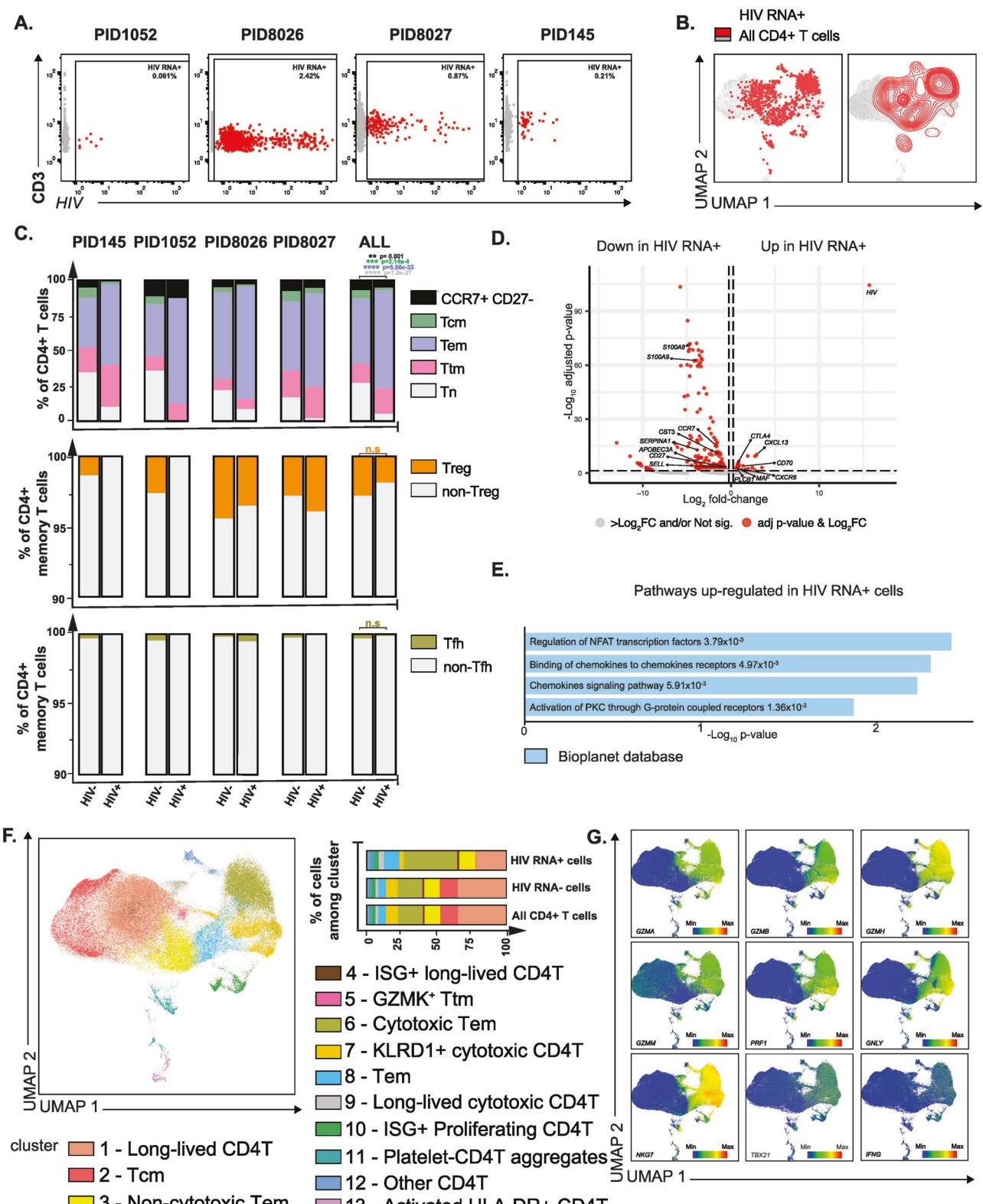

lineage markers, effector and cytotoxic profiles, and activation/proliferative states (Supplementary Fig. S10). HIV RNA+ cells were differentially distributed among the 13 clusters as compared to their uninfected counterparts, with highest enrichment of HIV RNA+ cells in cluster 6; 37% of all HIV RNA+ cells were located in this cluster, as compared to only 17.3% of total CD4 + T cells and 17.1% of HIV RNA- cells (Fig. 2F and Supplementary Table 6). Cluster 6 comprised mostly

Tem cells (Supplementary Fig. S9B), and was characterized by high expression of cytotoxic and cytolytic genes, including *GZMA, GZMB, GZMH, GZMM, PRF1, GNLY,* and *NKG7*, as well as the Th1-defining factors *IFNG* and *TBX21* (Fig. 2G and Supplementary Fig. 10). This finding suggests that cells from cluster 6 are Tem cells of a cytotoxic Th1 phenotype and is consistent with prior reports that HIV RNA+ cells from viremic PWH exhibit Th1 cytotoxic signatures [25,32].

**Fig. 2 | HIV RNA+ cells from viremic PWH are preferentially Tem cells and exhibit transcriptional signatures of cytolysis and cellular activation.**
**A** CD3 + CD4 + T cells transcribing HIV RNA (red dots) were identified by HIV-seq from 4 viremic PWH. HIV RNA- cells are depicted in gray. Percentages of HIV RNA+ cells are indicated in the upper right of each plot. **B** UMAP plots depicting HIV RNA+ cells as red dots (left) or contours (right) against a background of HIV RNA- CD4 + T cells (in gray) for all 4 individuals in panel A combined. **C** HIV RNA+ cells from viremic PWH are enriched in T effector memory (Tem) cells and disenriched in T central memory (Tcm) and naïve T (Tn) cells relative to HIV RNA- CD4 + T cells (top panel). The proportions of Treg vs. non-Treg (middle panel) and Tfh vs. non-Tfh cells (bottom panel) are not significantly different in HIV RNA+ vs. HIV RNA- memory CD4 + T cells. Statistical comparisons were performed using a generalized linear mixed-effects model. Pairwise comparisons tested differences in estimated marginal means of cell type membership, averaged over 4 PWH, using two-sided Z-tests without adjustment for multiple comparisons. **P < 0.01, ***P < 0.001,

****P < 0.0001. **D** Volcano plot displaying upregulated and downregulated transcripts in HIV RNA+ vs. HIV RNA- CD4 + T cells from 4 PWH, with select transcripts annotated. Red dots correspond to transcripts with fold-change expression ≥0.25log$_2$ and P < 0.05 (corrected for multiple comparisons), as determined by two-sided genewise quasi F-tests. **E** Pathway analysis comparing HIV RNA+ vs. HIV RNA- cells from 4 PWH using the Bioplanet 2019 database. P-values (to right of each pathway) were determined by a one-sided Fisher exact test. **F** Left: UMAP depicting the 13 clusters of CD4 + T cells from the viremic samples identified by Louvain clustering. Upper right: stacked graph showing the distribution of the different clusters among HIV RNA + CD4 + T cells, HIV RNA- CD4 + T cells, and total CD4 + T cells. **G**. Cells residing in the upper right of the UMAP, where cluster 6 cells localize, are defined by high expression of cytotoxic markers *GZMA, GZMB, GZMH, GZMM, PRF1, GNLY, NKG7*, and Th1-defining factors *IFNG* and *TBX21*. Heatmaps depict relative expression of the indicated transcripts. Source data are provided as a Source Data file.

Overall, our data demonstrate that HIV RNA+ cells from viremic PWH are heterogeneous but exhibit shared features, including being more likely to be Tem, and notably, displaying a previously undescribed state more conducive to viral replication, with low expression of restriction factors and increased activation of cellular pathways promoting HIV gene expression. Additionally, relative to their uninfected counterparts, they preferentially exhibit a cytotoxic signature characterized by higher expression of granzymes, perforin, and granulysin and a Th1 signature.

### HIV RNA+ cells from suppressed PWH are preferentially Tem cells but do not exhibit a cytotoxic signature

Three of the individuals we had analyzed in the context of active viremia (PID1052, PID8026, and PID8027) had specimens collected after > 24 weeks of ART suppression. Therefore, we next implemented a similar HIV-seq analysis pipeline to characterize these participant-matched HIV RNA+ cells in the context of ART suppression. We identified 14, 7, and 4 HIV+ cells out of 15,468, 42,618, and 17,072 CD4 + T cells for PID1052, PID8026 and PID8027, respectively. A total of 25 HIV RNA+ cells were detected out of the 75,158 CD4 + T cells analyzed, corresponding to an HIV RNA+ cell frequency of 0.016% to 0.091% (Fig. 3A). As for the viremic specimens, HIV RNA+ cells were broadly distributed among CD4 + T cells, demonstrating heterogeneity (Fig. 3B), and were enriched among Tem cells (Fig. 3C). In fact, in two individuals (PID8026 and PID8027), the entire HIV RNA+ cell population was exclusively found within Tem cells. When comparing HIV RNA+ and HIV RNA- cells, we found no DEPs that were consistent across study participants (Supplementary Fig. 11). Likewise, few DEGs were observed between the HIV RNA+ and HIV RNA- CD4 + T cells, with the notable exception of a handful of downregulated genes (Fig. 3D and Supplementary Table 7). These included the Tcm marker *SELL*, the alarmin *S100A9*, and the cysteine protease inhibitor *CST3*, all of which were also preferentially downregulated among HIV RNA+ cells during active viremia (Fig. 2).

Louvain clustering analysis revealed that unlike HIV RNA+ cells collected during viremia, those collected during suppression were not enriched among cluster 6 (cytotoxic Tem). Instead, the cluster distribution of the infected cells under ART revealed that HIV RNA+ cells were primarily distributed among cluster 1, with 52% of all HIV RNA+ cells being in cluster 1 as compared to only 28.1% in both total CD4 + T cells and HIV RNA- cells (Fig. 3E and Supplementary Table 6). Cluster 1 comprises CD4 + T cells expressing high levels of the *IL7R* transcript (Supplementary Fig. 10), which encodes for a receptor that is characteristic of long-lived cells and is important for homeostatic proliferation. Interestingly, cluster 1 also includes cells with Th2 features as defined by high *GATA3* expression (Supplementary Fig. 10), although *IL4* and *IL5* – cytokines characteristic of Th2 cells – were not detected.

We also observed a small enrichment of infected cells in cluster 3, with 16% of HIV RNA+ cells located in this cluster, compared with 9.7% in both total CD4 + T cells and HIV RNA- cells. Like cluster 1, cluster 3 also highly expressed *IL7R* (Supplementary Fig. 10). Interestingly, *MALAT1*, a long non-coding RNA (lncRNA) shown to promote HIV-1 transcription and infection[45], was also highly expressed in this cluster (Supplementary Fig. 10).

Our observation that cells in both clusters 1 and 3 express high levels of *IL7R* supports the notion that the active HIV reservoir persists in long-lived/stem-like cells with proliferative potential. Indeed, IL7R expression – at both the transcript and protein levels – was spatially localized to regions of the UMAP occupied by clusters 1 and 3 (Fig. 3F). Furthermore, when we directly compared the HIV RNA+ cells to HIV RNA- cells, we found that *IL7R* transcript and IL7R protein were both more highly expressed on the HIV RNA+ cells (*P* < 0.05; Fig. 3G). That HIV RNA+ cells during ART persist in stem-like cells is further supported by our observation that a substantial proportion (44%) of the HIV RNA+ cells express *BCL-2*, a pro-survival, anti-apoptotic gene implicated in driving long-term persistence of HIV[46–49] (Fig. 3H). Taken together, these findings suggest that in blood, most HIV RNA+ cells during ART suppression reside not within cytotoxic CD4 + T cells – as was observed during viremia – but rather within a long-lived population of memory CD4 + T cells.

### CD4 + T cells exhibit stronger antiviral immunity and upregulate the integrated stress response pathway during viremia as compared to after ART suppression

Leveraging the fact that HIV-seq was performed on paired specimens from before vs. after ART suppression, we then compared the transcriptomes of total CD4 + T cells across these two conditions. Total CD4 + T cells coming from viremic vs. suppressed specimens were transcriptionally divergent, as reflected by distinct UMAP localizations (Fig. 4A, Supplementary Fig. 12), even though their distribution among classical CD4 + T cell subsets was similar (Fig. 4B). Although DEGs were identified (Fig. 4C, Supplementary Table 8), none retained statistical significance following correction for multiple comparisons. As an exploratory analysis, however, we assessed the DEGs with the lowest raw p-values to gain insights into potential cellular pathways distinguishing CD4 + T cells during active viremia from those following ART suppression. This analysis revealed that during viremia, CD4 + T cells preferentially express higher levels of cytotoxic and proinflammatory genes (*SP100, IL6*), and genes related to interferon α, β and γ signaling (*IFI27, IFI44L, IFIT3, IFI6, IFI44, ISG15, IFITM10, IFI3O, IFI27L1, MX1, IFIH1, IFNLR1, IFIT5, NEAT1, TYMP*). Some of these (*IFI44L, ISG15, NEAT1, TYMP*) have previously been reported to be higher among CD4 + T cells during active viremia as compared to ART treatment[50]. This upregulation may reflect an antiviral host response triggered by high levels of viral transcripts and proteins during viremia. Other highly expressed genes

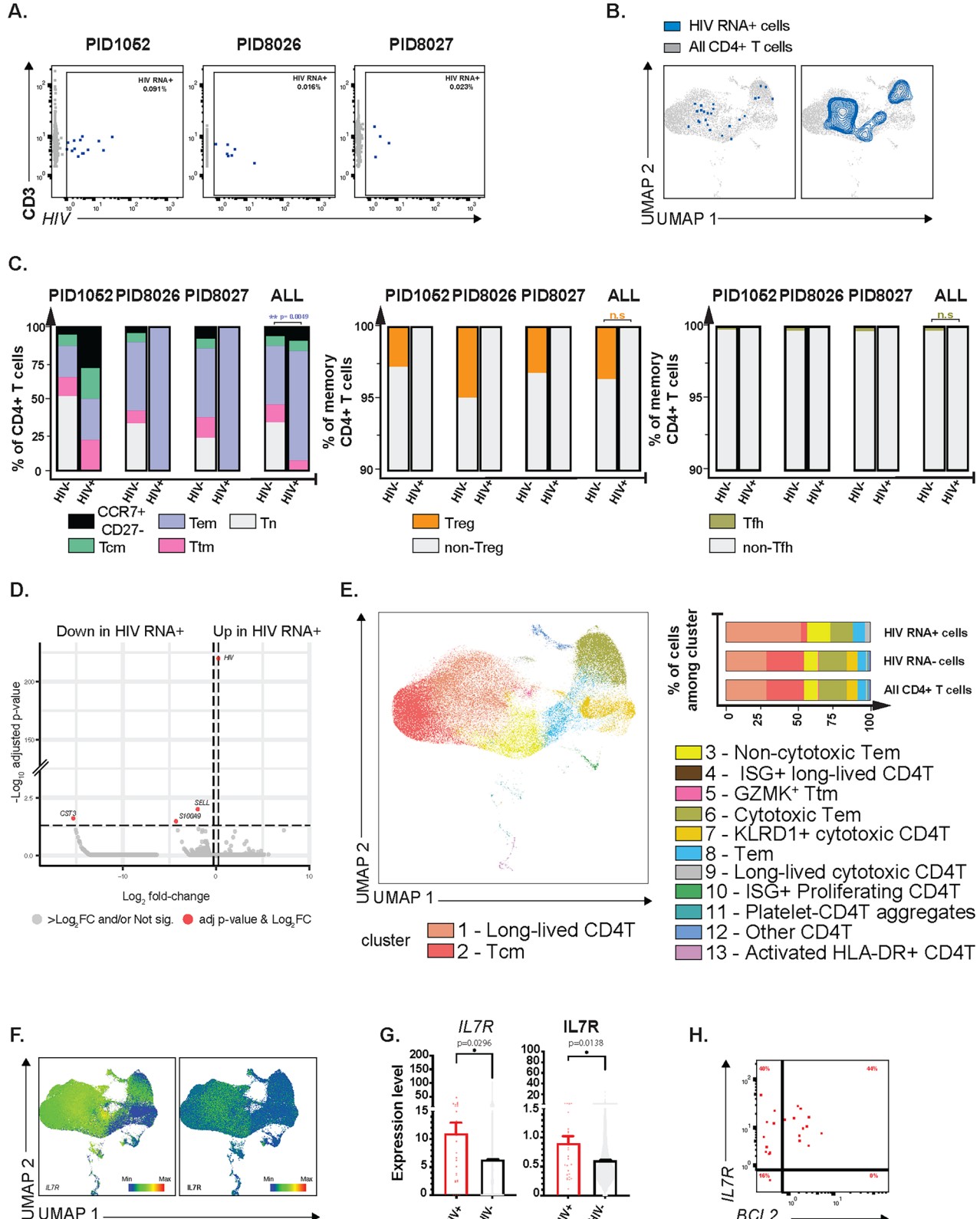

during viremia included cytotoxic genes (*GZMB, GZMK*) whose expression can be induced by cytokines, such as IL2 and IL15[51], and during inflammation and viral infections [52].

Also of interest was that during viremia, CD4 + T cells increased expression of *EIF2AK2* (Fig. 4C, Supplementary Table 8), a key gene involved in integrated stress response (ISR), a pathway previously reported to be induced during acute HIV infection[53,54]. During ISR,

*EIF2AK2* expression is induced by type I IFNs, which then upon binding to viral dsRNA can initiate a cascade of events culminating in diminished protein translation. Indeed, when we performed DEG analysis of all donors combined using an approach previously implemented to identify DEGs between total CD4 + T cells during active HIV viremia vs. after ART suppression[25,50] (see Methods), we observed downregulation of numerous ribosomal transcripts (RPL and RPS transcripts) during

**Fig. 3 | Most HIV RNA+ cells from ART-suppressed PWH exhibit stem-like rather than cytotoxic features. A** CD3 + CD4 + T cells expressing HIV RNA (blue dots) were identified by HIV-seq from 3 ART-suppressed PWH. HIV RNA- cells are depicted in gray. Percentages of HIV RNA+ cells are indicated in the upper right. **B** UMAP plots depicting HIV RNA+ cells as blue dots (left) or contours (right) against a background of HIV RNA- CD4 + T cells (in gray) for all 3 PWH from panel A combined. **C** Left panel: distribution of Tn, Tcm, Tem, Ttm, and memory CCR7+ CD27- cells among HIV RNA- and HIV RNA + CD4 + T cells. Remaining panels: proportions of Treg and non-Tregs, and Tfh and non-Tfh, among HIV RNA- and HIV RNA+ memory CD4 + T cells. Statistical comparisons were performed using a generalized linear mixed-effects model. Pairwise comparisons tested differences in estimated marginal means of cell type membership, averaged over 3 PWH, using two-sided Z-tests without adjustment for multiple comparisons. \*\*$P < 0.01$. **D** Volcano plot displaying differentially expressed transcripts in HIV RNA+ vs. HIV RNA- CD4 + T cells from 3 ART-suppressed PWH, with select transcripts annotated.

Red dots indicate transcripts with ≥0.25log$_2$ fold-change expression and p value < 0.05 (adjusted for multiple comparisons), as determined by a two-sided genewise quasi F-tests. **E** Left: UMAP depicting the 13 clusters of CD4 + T cells from the ART-suppressed PWH identified by Louvain clustering. Upper right: stacked graph showing the distribution of the different clusters among HIV RNA + CD4 + T cells, HIV RNA- CD4 + T cells, and total CD4 + T cells. **F** UMAPs depicting expression of *IL7R* transcript (left) and IL7R protein (right) among CD4 + T cells from 3 PWH. **G** HIV RNA+ cells (red dots; *n* = 25 from 3 PWH) express significantly higher levels of *IL7R* transcript (left) and IL7R protein (right) relative to HIV RNA- cells (*n* = 74,913 from 3 PWH; gray violin plots). Error bars represent standard errors of the mean. P values were determined by a two-sided Wilcoxon rank-sum test with Benjamini-Hochberg correction. **H** Expression of *IL7R* and *BCL-2* transcripts in HIV RNA + CD4 + T cells (red dots). Percentages of HIV RNA+ cells CD4 + T cells are indicated in each quadrant. Source data are provided as a Source Data file.

viremia (Supplementary Fig. 13 and Supplementary Table 9). The diminished ribosomal transcript levels were accompanied by diminished expression of *EEF2*, whose downregulation has been shown to reduce active ribosomes[54] and shut down mRNA translation, resulting in overall diminished viral protein synthesis[55]. Hence, the ISR pathway, which serves to coordinate cellular responses to various stressors by regulating protein synthesis and gene expression and is a host response to limit viral spread, appears to be a characteristic feature of acute untreated HIV infection that is turned off upon ART suppression.

### Relative to HIV RNA+ cells during viremia, those during ART suppression upregulate TGFβ signaling pathways and exhibit diminished activation of host responses

We then compared the HIV RNA + CD4 + T cells from the paired viremic and ART-suppressed timepoints. Classical CD4 + T cell subset distributions of HIV RNA+ cells were similar between the two timepoints, with preferential distribution among Tem cells for both (Fig. 4D). DEP analysis revealed lower CD4 expression on HIV RNA+ cells during viremia as compared to during ART suppression (Supplementary Fig. 14), perhaps due to elevated expression of Nef, an HIV accessory gene that potently downregulates cell-surface CD4 expression in productively-infected cells[56]. Multiple DEGs were identified between the timepoints, but none retained statistical significance after correction for multiple comparisons. As an exploratory analysis, we assessed the identities of the DEGs based on the raw p-values (Fig. 4E and Supplementary Table 10). This analysis revealed upregulation of multiple ISGs and proinflammatory genes in HIV RNA+ cells during viremia compared to suppression, including *IFI27*, *MX1*, *ISG15*, *DUSP2*, and *SP100*. These genes were also upregulated among total CD4 + T cells during viremia (Fig. 4C and Supplementary Table 8). *IFI27* in particular was increased by more than 10-fold in HIV RNA + CD4 + T cells during viremia as compared to those during ART. Given that HIV-1 Vpr and Tat can directly induce *IFI27* production[57–59], and that expression of these viral proteins may be diminished in the context of ART suppression due to multiple blocks to HIV transcription[7–9], it is possible that elevated expression of Vpr and Tat is driving *IFI27* expression prior to ART initiation. Interestingly, *IFI27* levels correlate with inflammation and disease progression during both HIV-1 and HIV-2 infection[60,61], and they may contribute to HIV pathogenesis [60,61].

DEGs associated with HIV transcription were also upregulated in HIV RNA+ cells during viremia as compared to after ART suppression. HIV RNA+ cells during viremia expressed higher levels of *SRRM1*, a modulator of HIV-1 splicing that is involved in the regulation of Tat and Nef expression[62], and lower levels of two genes implicated in silencing HIV transcription and translation: *RBL2*, which recruits and targets histone methyltransferases, leading to epigenetic transcriptional repression; and *RPS10*, which binds HIV Nef to form a complex that decreases viral protein synthesis[63]. Together, these findings suggest

that during viremia, HIV RNA+ cells exhibit a transcriptional profile that favors the production of more HIV transcripts. This finding accords with our having observed elevated HIV transcript levels among HIV RNA+ cells during viremia as compared to during suppression in two out of our three participants (11- and 14-fold increase, respectively, for PID8026 and PID8027).

Pathway analysis of the DEGs in HIV RNA+ cells before versus after ART suppression supported the notion that HIV RNA+ cells during viremia are preferentially in an activated, antiviral state, characterized by upregulation of multiple interferon pathways (IFN I and IFN II) (Fig. 4F, top). Conversely, HIV RNA+ cells during ART suppression preferentially upregulated TGF-β signaling (Fig. 4F, bottom). This finding was driven by upregulation of *RBL2* and *ITGB1* (Supplementary Table 10), genes associated with the TGF-β signaling pathway. *ITGB1*- and *RBL2*-associated TGF-β activation has been implicated in tumor suppression and cancer growth arrest[64,65]. Although *ITGB1* and *RBL2* have not been directly implicated in HIV infection, TGF-β signaling has been recently implicated in promoting HIV latency[66,67]. Our data suggest that this TGF-β-associated signature is a phenomenon that only emerges in the context of ART suppression, as HIV RNA+ cells during viremia do not exhibit such a signature. Overall, these results indicate that the transcriptional profiles of HIV RNA+ cells differ depending on whether or not ART is present, and that the features of HIV RNA+ cells during viremia cannot be assumed to be the same as those during ART suppression.

## Discussion

In this study, we introduce HIV-seq as a method to improve the efficiency of capturing HIV transcripts during scRNA-seq. Applying HIV-seq to blood samples collected at viremic and suppressed timepoints from the same set of individuals allowed us to confirm previous findings and discover new features of HIV-transcribing cells in viremic individuals, and to discern differences between these cells and active reservoir cells that persist during stable ART suppression.

HIV-seq increased detection of HIV transcripts from PWH and facilitated in-depth scRNA-seq of the highest numbers of HIV RNA+ cells analyzed to date, in the context of both viremia and ART suppression. We recovered and analyzed 1,072 HIV RNA+ cells from four viremic PWH, in comparison to prior methods using classic scRNA-seq[32], ECCITEseq[25], and DOGMAseq[50], which had characterized 164 total cells from 14 PWH (with one highly viremic donor further sequenced to gain an additional 223 HIV RNA+ cells), 81 total cells from 6 PWH, and 256 total cells from 6 PWH, respectively. We note, however, that >1000 HIV RNA+ cells were detected by scRNA-seq from a single participant in one study reporting the negative association between transcript levels of the restriction factor *PTMA* and HIV transcript levels[32]. Our identification of 25 HIV RNA+ cells from three people on ART is also higher than prior studies, which had reported two total cells from 14 people[32], 9 total cells from 6 people[25], and 14

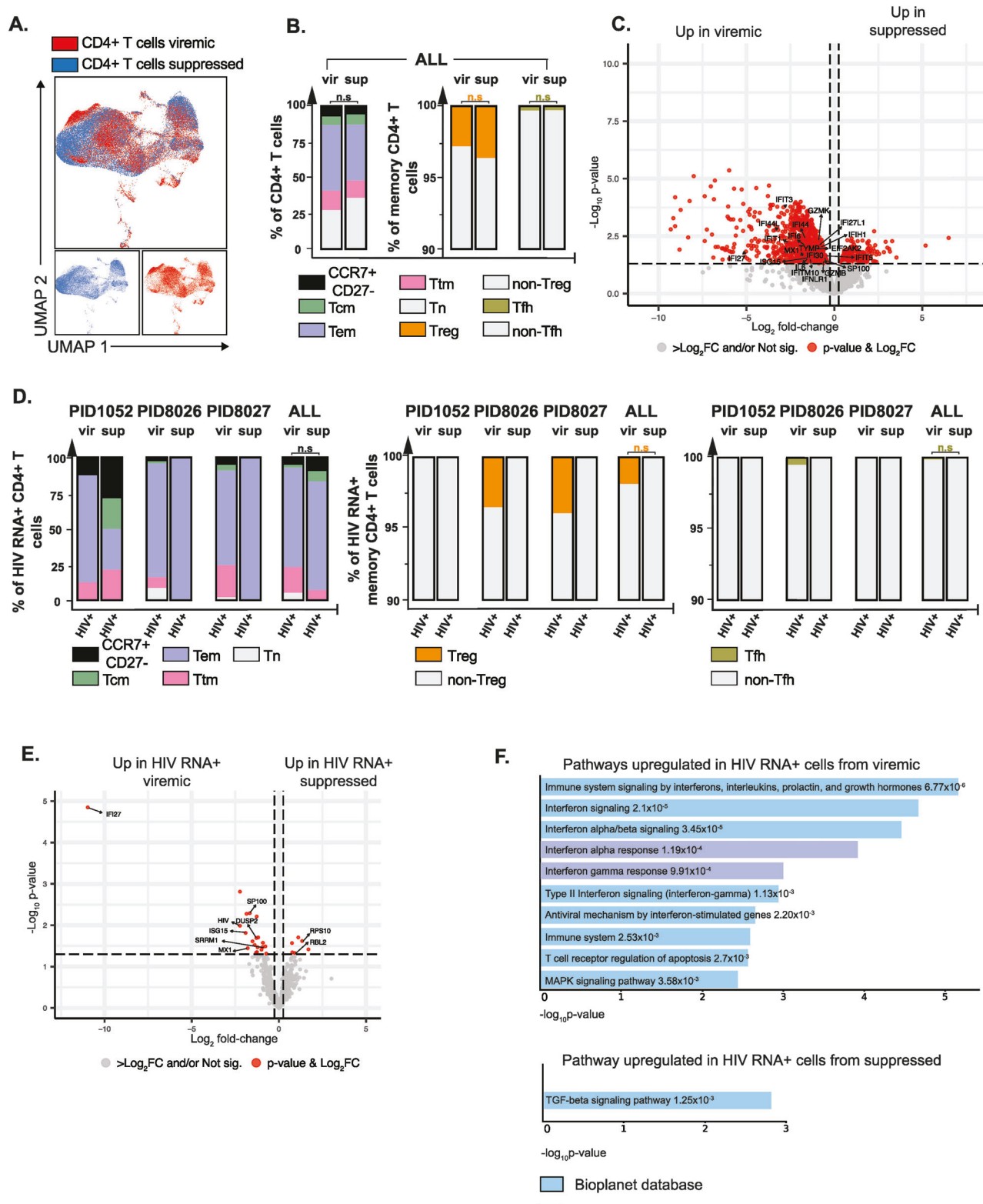

total cells from six people[50]. The higher number of HIV RNA+ cells identified in our study may reflect differences in HIV RNA capture and/or the underlying frequency of HIV RNA+ cells.

The elevated numbers of HIV reads identified by HIV-seq also enabled an in-depth analysis of the distribution HIV transcripts among infected cells. We found reads spanning the entire HIV genome, regardless of whether we included HIV capture primers. During viremia, the most frequently detected transcript was *pol*, followed by *gag*,

*vif*, *env*, and *tat*. Likewise, *gag* and *pol* were most common during ART suppression. This distribution likely reflects three key factors. First, the technology used for scRNA-seq can introduce biases at the stages of RNA capture (use of poly-dT +/- specific sequences), RT, binding of template switch oligo (to CCC trinucleotides), second strand synthesis, amplification (amplicon size), fragmentation, ligation, sequencing, and analysis (location of cell/transcript barcodes). Second, alignment efficiency is impacted by the extent to which a sequenced read matches

**Fig. 4 | HIV RNA+ cells from viremic PWH exhibit a pro-inflammatory and anti-viral state while those from ART-suppressed PWH exhibit properties that favor senescence and HIV restriction. A** UMAP plots of total CD4 + T cells from viremic (red) vs. suppressed (blue) timepoints from PID1052, PID8026 and PID8027 combined. **B** Bar graphs depicting distribution of Tn, Tcm, Tem, Ttm, memory CCR7 + CD27- cells, as well as the distribution of Treg, non-Treg, Tfh, and non-Tfh, during viremia (vir) and ART suppression (sup). Results from all 3 PWH are combined. Differences were not significant (n.s.) as determined by generalized linear mixed model (GLMM). Pairwise comparisons tested differences in estimated marginal means of cell type membership, averaged over n = 3 PWH, using two-sided Z-tests without adjustment for multiple comparisons. **C** Volcano plot displaying differentially expressed transcripts in CD4 + T cells from viremic vs. suppressed time points, with select transcripts annotated. Red dots correspond to transcripts with 0.25log$_2$ fold-change expression and with a non-adjusted p value < 0.05, as determined by a two-sided gene-wise quasi F-tests. **D** Bar graphs depicting distribution of Tn, Tcm, Tem, Ttm, and memory CCR7 + CD27- cells, as well as the distribution of Treg, non-Treg, Tfh, and non-Tfh, among HIV RNA+ memory CD4 + T cells during viremia (vir) and upon ART suppression (sup). Results from all 3 PWH are combined. Differences were not significant (n.s.) as determined by GLMM. Pairwise comparisons tested differences in estimated marginal means of cell type membership, averaged over n = 3 PWH, using two-sided Z-tests without adjustment for multiple comparisons. **E** Volcano plot displaying up- and down-regulated transcripts in HIV RNA+ cells from 3 PWH during viremia vs. ART suppression, with select transcripts annotated. Red dots correspond to transcripts with 0.25log$_2$ fold-change expression and with a non-adjusted p value < 0.05, as determined by a two-sided gene-wise quasi F-tests. **F** Pathway analysis comparing HIV RNA+ cells from 3 PWH in the context of viremia vs. suppression, using the Bioplanet 2019 and MSigDB hallmark databases. P-values (shown next to the corresponding pathway) were calculated using a one-sided Fisher's exact test. Source data are provided as a Source Data file.

our subtype B consensus sequence. *Gag* and *pol* genes are the most conserved among subtype B HIV-1 isolates[68,69], which may have contributed to their being the most frequently identified HIV transcripts in our scRNA-seq analysis. Lastly, read distribution can be affected by the relative proportions of each transcript type. In particular, blocks to HIV transcription in active reservoir cells primarily occur before the *env/nef* regions[7], resulting in a higher presence of 5' reads in the samples. All these factors together may have accounted for our preferential identification of *gag* and *pol* transcripts, irrespective of HIV-seq.

We identified over a thousand HIV RNA+ cells by HIV-seq in the absence of ART, enabling an in-depth analysis of the features of HIV-infected cells during viremia. This analysis both confirmed prior studies and also revealed new insights not previously reported. HIV RNA+ cells during viremia were predominantly of the Tem phenotype, consistent with previous reports[22,25]. One likely explanation is the increased susceptibility of Tem cells to HIV infection relative to their Tcm counterparts[24]. Also consistent with prior studies[25,32,50] is our observation that HIV RNA+ cells preferentially reside in a cluster of cells exhibiting a Th1-like effector/cytotoxic phenotype, defined by preferential expression of Th1-defining markers along with cytotoxic markers including granzymes, perforin, and granulysin. Active viral replication may promote the acquisition and maintenance of cytotoxic functions by CD4 + T cells by eliciting a sustained pro-inflammatory environment, which has been shown in the context of cancer[70] as well as during influenza infection [71].

We also discovered that during active viremia, HIV RNA+ cells expressed lower levels of the restriction factors *APOBEC3A* and *SERPINA1* compared to their HIV RNA- counterparts, which to our knowledge has not previously been reported and may reflect an immune evasion mechanism mediated by the virus. APOBEC3A is a DNA cytidine deaminase that exhibits antiviral activity, including against HIV[72,73], and can also help maintain HIV latency in CD4 + T cells through recruitment of epigenetic silencing machinery to the LTR[74]. The HIV accessory gene *vif*, however, can target APOBEC3A for proteasome-mediated degradation[72]. Our finding of decreased *APOBEC3A* expression at the transcript level suggests that there may be mechanisms beyond proteasome-mediated degradation to suppress APOBEC3A activity. SERPINA1 is a restriction factor that is induced during inflammation and inhibits HIV LTR-driven transcription[75], and SERPINA1 expression can be regulated by methylation, independently of inflammation[76]. The extent to which HIV RNA+ cells downregulate SERPINA1 expression through methylation is unknown, but this mechanism is conceivable given the profound epigenetic changes induced by HIV infection[77]. Hence, diminished expression of both *APOBEC3A* and *SERPINA1* in HIV RNA+ cells during acute viremia may promote rapid production of new virions by promoting HIV transcription.

In addition to analyzing HIV RNA+ cells, we also leveraged our in-depth sequencing datasets to assess the extent to which active HIV viremia affects the transcriptomes of total CD4 + T cells. Comparing the scRNA-seq profiles of total CD4 + T cells before and after ART suppression revealed an upregulation of proinflammatory genes and genes related to interferon α, β, and γ signaling during viremia. This finding aligns with a recent demonstration of elevated type I IFN gene expression (*IFI44L, ISG15, XAF1, NEAT1, TYMP, TRIM22*) in total CD4 + T cells during untreated HIV infection[50], and is consistent with the notion of a pro-inflammatory response induced by active viral replication.

Unexpectedly, we also discovered that total CD4 + T cells during viremia upregulated the ISR pathway relative to their counterparts during ART suppression. In general, ISR serves as a protective host response against viruses by reducing global protein synthesis to inhibit viral replication, and in some cases can further induce apoptosis of infected cells[53]. However, some viruses – including HIV – seem to have evolved mechanisms to hijack or benefit from ISR[53,78]. For example, the ISR-associated transcription factor ATF4 can bind to the HIV promoter to stimulate HIV transcription[78]. It is thus possible that global upregulation of ISR among CD4 + T cells creates an intracellular environment favorable for HIV gene expression, thereby facilitating rapid systemic spread of the virus during untreated infection.

In addition to revealing insights into HIV pathogenesis and spread during untreated infection, HIV-seq also allowed for a more focused analysis of active reservoir cells in aviremic individuals. To our knowledge, no previous study has applied scRNA-seq to specifically analyze HIV RNA+ cells from the peripheral blood of ART-suppressed PWH. Prior studies have either combined HIV RNA+ cells from viremic and suppressed samples—likely due to low numbers of detectable HIV RNA+ cells during ART[25,32] —or have focused their primary analysis on HIV RNA+ cells during viremia [50].

Importantly but perhaps not surprisingly, we found that HIV RNA+ cells during ART suppression differ from those during active viremia, suggesting that HIV RNA+ cells during viremia may not fully represent the features of reservoir cells that persist during ART suppression. Although HIV RNA+ cells in both instances were predominantly Tem, those during ART suppression did not preferentially harbor a cytotoxic phenotype. This finding can be explained by the general short-lived nature of effector/cytotoxic lymphocytes, which has been described for CD8+ CTLs[79]. By contrast, HIV reservoir cells are long-lived, and recent studies of total and genome-intact HIV reservoir cells have suggested preferential expression of markers of cell survival[80,81]. Although it may seem perplexing that we observed active reservoir cells to preferentially reside among Tem, which are generally considered short-lived[82], it is worth noting that long-lived Tem cells have been described in the context of viral infections [83–85].

Supporting the notion that active reservoir cells, like the total reservoir cells, exhibit features of longevity, we found that during ART suppression, HIV RNA+ cells preferentially resided in two clusters of

cells expressing high levels of CD127, the alpha chain of the IL7 receptor, which is a major driver of homeostatic proliferation. IL7 can promote stabilization of a long-lived reservoir of HIV-infected cells[86] and is associated with a slower contraction of the total HIV reservoir (as defined by HIV DNA levels) over time[87]. These observations together suggest that IL7 may also drive persistence of the active reservoir. We also found that a substantial proportion of reservoir cells express BCL-2, an anti-apoptotic protein that inhibits apoptosis by regulating mitochondrial membrane permeability and preventing the release of pro-apoptotic factors[88]. Interestingly, ex vivo treatment of cells from ART-suppressed PWH with different BCL-2 inhibitors, such as venetoclax[46] or obatoclax[49] decreases the pool of genome-intact HIV-infected cells. Furthermore, venetoclax delays viral rebound upon ART interruption in a humanized mouse model of HIV persistence[46]. Our data suggest that a considerable fraction of the active reservoir, like genome-intact reservoir cells, should also be sensitive to BCL-2 inhibitors. The outcome of a recently initiated clinical trial testing venetoclax as a therapeutic approach to achieve HIV remission in ART-suppressed PWH[89] will be interesting in that regard.

Besides exhibiting stem-like properties, active reservoir cells also exhibited increased activation of the TGF-β pathway. Recent studies in a non-human primate model of SIV infection implicated TGF-β in promoting HIV persistence, through mechanisms related to both the cytokine's immunosuppressive effects as well as its ability to suppress viral gene expression[66,67]. Our finding that active reservoir cells preferentially activate the TGF-β pathway – in particular by upregulating ITGB1, which mediates release of the active form of TGF-β[64], and RBL2, which in response to TGF-β activation mediates changes cell cycle progression[65] – suggests that persistent HIV may utilize the TGF-β pathway to achieve immune evasion. Immune evasion may be particularly important for active reservoir cells, as these cells may express HIV proteins that can then be processed for recognition by HIV-specific CD8 + T cells[90–92]. Our finding that active reservoir cells utilize the TGF-β pathway, alongside evidence that galunisertib (a TGF-β1 receptor inhibitor) promotes ex vivo reactivation of HIV from cells of ART-suppressed PWH[66], favors the notion that targeting the TGF-β pathway may be a viable approach for the "kick and kill" strategy for eliminating HIV reservoir cells.

Together, these results demonstrate that HIV RNA+ cells during active viremia are primarily cytotoxic CD4 + T cells with diminished expression of restriction factors targeting HIV transcription, while those during ART suppression exhibit features consistent with long-term survival, including anti-apoptotic and homeostatic proliferation mechanisms, along with potential involvement of TGF-β signaling pathways to achieve immune evasion. Our data also suggest a functional switch in immune signaling between viremic and virally-suppressed states, whereby infected cells upon ART suppression adopt a more anti-inflammatory, tolerogenic phenotype. This shift may enable reservoir persistence by dampening host antiviral responses and engaging pathways that promote immune evasion and long-term survival. Future studies could apply HIV-seq to more broadly characterize the active reservoir, for example in the context of tissues where HIV primarily persists. In addition, applying HIV-seq in the context of clonal expansion analysis using single-cell VDJ analysis, and further developing this technique to allow for multiplexing with other platforms (e.g. single-cell ATAC-seq), will have utility in furthering our understanding of the mechanisms by which HIV can persist long-term despite ART suppression of viremia.

## Methods
### Ethics statement
The study was approved by the Committee on Human Research (CHR), the Institutional Review Board for the University of California, San Francisco (approval #11–07551 and #10-01561). All study participants provided written informed consent.

### Study Population
The study participants were 4 people living with HIV (median age = 38.5; median CD4 count = 490 cells/mm$^3$; Table 1). Participants were selected based on availability of cryopreserved PBMC from before the start of ART (n = 4), with a preference for those with an additional longitudinal sample of cryopreserved PBMC obtained after ART suppression (n = 3). Paired and longitudinal, archived PBMC samples were obtained from the UCSF Treat Acute Study and the SCOPE cohort. Samples were collected prior to ART initiation (Week 0) and following ART suppression (Week 24: PID8026; Week 45: PID8027; and Week 70: PID1052). A total of two aliquots of $10^7$ cells each were obtained for each time point (viremic and suppressed), with one aliquot preserved for HIV DNA/RNA measurements. An additional participant (PID0145), for whom only the viremic (Week 0) time point was available, was recruited from the San Francisco VA Medical Center. Three additional PBMC samples from HIV-uninfected donors were used as negative controls, to establish the extent to which HIV-seq results in false-positive identification of HIV reads and to set up the HIV transcript threshold for identification of HIV-infected cells.

In this pilot study where it was only feasible to study a small number of participants, and given the difficulty in finding paired longitudinal samples from before and after ART, we accepted the first available samples, which were provided to us in de-identified form. Therefore, sex/gender was not considered in the study design. Data on sex/gender was obtained after completion of data analysis, and was self-reported. With only three male participants and one female participant, the numbers were deemed insufficient for a post-hoc analysis.

### HIV DNA measurements
For each participant, total HIV DNA (R-U5-pre-*gag* region) levels were measured to ascertain whether their reservoir was sufficiently high for detection with limited cell inputs (up to 10,000 cells for each well of the 10X Genomics scRNA-seq platform). An aliquot of $10^7$ cryopreserved cells was tested from each time point (viremic and suppressed) for each individual. Each aliquot was thawed, cryopreservation medium was removed, and total RNA and DNA were extracted using TRI Reagent (Molecular Research Center, Inc., Cincinnati, OH) as per manufacturer's instructions, with the following modifications: poly-acryl carrier (Molecular Research Center, Inc., Cincinnati, OH) was added to TRI reagent prior to lysis, RNA was resuspended in RNase free-water, DNA was extracted using back extraction buffer (4 M guanidine thiocyanate, 50 mM sodium citrate, 1 M Tris), polyacryl carrier was added to the aqueous phase containing the DNA, and DNA was resuspended in QIAGEN buffer EB[8]. Cellular DNA was fragmented using a QIAshredder column (QIAGEN) to facilitate incorporation into droplets.

HIV DNA and copies of the housekeeping gene *TERT* (telomere reverse transcriptase) were measured in duplicate by droplet digital (dd)PCR (Bio-Rad QX200)[7]. Each reaction consisted of a 20 μL solution. The ddPCR for *TERT* contained 10 μL of ddPCR Probe Supermix (no dUTP) [Bio-Rad], 1uL of 20x TaqMan™ Copy Number Reference Assay, human, TERT (Thermo Fisher Scientific), and 50 ng of cellular DNA. The ddPCR for HIV DNA contained 10 μL of ddPCR Probe Supermix (no dUTP) [BioRad], 900 nM of each primer (F: gcctcaataaagcttgccttga; R: gggcgccactgctagaga; both from Thermo Fisher Scientific), 250 nM of probe (5'FAM-ccagagtcacacaacagacgggcaca-3' nonfluorescent quencher/Minor Groove Binder [MGB]; Thermo Fisher Scientific), and 500 ng of cellular DNA. Droplets were amplified with the following cycling conditions: 10 min at 95 °C, 45 cycles of 30 s at 95 °C and 59 °C for 60 s, and a final droplet cure step of 10 min at 98 °C. Droplets were read and analyzed using the QuantaSoft software in the "Absolute" quantification mode. A threshold of three HIV DNA copies/10,000 cells at the suppressed time point was set as a threshold for study inclusion, as this infection frequency was predicted to yield a reasonable likelihood of detecting at least one HIV RNA+ cell using the 10X Genomics

**Table 1 | Characteristics of HIV+ study participants**

| Participant ID | Cohort | Gender | Race/ Ethnicity | Age | CD4 count week 0 (cells/mm³) | VL at Week 0 | VL at Week 24 or 45 | Drug Regimen | Samples |
|---|---|---|---|---|---|---|---|---|---|
| 8026 | SCOPE | Male | Mixed - Latino, Native American | 41-50 | 366 | >1,000,000 | <40 | FTC/TAF, DTG | PBMC (20 ×10⁶) |
| 1052 | Treat Acute | Female | White/European American | 21-30 | 933 | <500 | <40 | 3TC, TDF, ATV, RTV ABC/3TC, ATV, RTV | PBMC (20 ×10⁶) |
| 8027 | SCOPE | Male | White | 31-40 | 590 | >1,000,000 | <40 | FTC/TDF, DTG | PBMC (20 ×10⁶) |
| 0145 | SFVA | Male | Latino | 31-40 | 390 | >850,000 | N/A | N/A | PBMC (15 ×10⁶) |

scRNA-seq platform. All samples subjected to single-cell sequencing in this study met this threshold.

### scRNA-seq sample preparation

**PBMC.** A total of $10^7$ PBMCs from each donor were thawed at 37 °C and washed once with warm media (RPMI [Corning Inc., Corning, NY] supplemented with 10% FBS [VWR, Radnor, PA]) and then resuspended in FACS buffer (RPMI, supplemented with 2% FBS and 2 mM EDTA [Thermo Fisher Scientific]) prior to counting. After setting aside $10^5$ cells from each sample for downstream PBMC spike-in, CD4 + T cells were purified from the remaining cells by negative selection using the EasySep Human CD4 + T Cell Enrichment Kit (StemCell). The PBMCs from the same donor that were set aside were then spiked back in at a ratio of 5:100 in order have non-CD4 + T cells to help establish gates for defining HIV RNA+ cells.

**TotalSeq-C Antibody Staining.** TotalSeq-C pooled antibody mix (Biolegend) was prepared so as to contain 0.4 µg of each antibody in a final volume of 100 µL/sample, in PBS (Ca + + and Mg+ free, Corning Inc.) containing 3% FBS, as per the devised panel (Supplementary Table 1). A million cells were pelleted, resuspended in RPMI supplemented with 3% FBS, and incubated at 4 °C for 10 min with Fc block (Miltenyi) at a 1:10 dilution. 100 µL/sample of TotalSeq pooled antibody mix was then added to the cells, which were incubated for 30 min at 4 °C. Cells were then washed three times in RPMI containing 0.04% BSA, strained using a 40 µm cell filter (BD Falcon), and resuspended at a concentration of 1000 cells/µL. The TotalSeq-C panel was used to facilitate manual gating: CD45 to identify immune cells; CD19, CD20, CD14, and CD8 to exclude B cells, myeloid cells, and CD8 + T cells; CD3 and CD4 to identify CD4 + T cells; CD45RA and CD45RO to distinguish between naïve and memory CD4 + T cells; and CD197 (CCR7), CD27, CD62L, CD127, CD279 (PD1), and CD49d as additional subset/phenotype markers.

### Custom HIV primers

Custom HIV-seq primers (obtained from Thermo Fisher Scientific) were designed by appending HIV-specific capture sequences (Supplementary Table 2) to a non-poly(dT) PCR handle (Fig. 1). The concentration of 10X Poly(dT) oligo (poly-dT RT Primer PN 2000007) is estimated to be 2.7 µM (100pmol in 36.8 µl of RT mix) in each 10X RT reaction[93]. Initial experiments spiking in different concentrations of pre-*gag* primers revealed that 0.67 µM of primer resulted in a significantly lower total cDNA yield as compared to the 0.33 µM condition and the condition where no HIV primers were added (Supplementary Fig. 1). Hence, we implemented 0.33 µM of the primers moving forward. The primer pool (containing 84 nM of each primer) was added to the 10X Genomics RT reaction mix containing RT Reagent B, Poly-dT RT Primer, Reducing Agent B, and RT Enzyme C (HIV-Seq RT Mix), which was used for RT of encapsulated cells as described below.

### RT, library preparation and sequencing

Chromium Next GEM Single Cell 5' Reagent Kits v2 (Dual Index) with Feature Barcode technology for Cell Surface Protein & Immune Receptor Mapping (10X Genomics, PN1000263) and the Chromium™ Controller (10X Genomics, PN120223) were used for gene expression library and cell-surface protein library generation. Briefly, 20 µl of a TotalSeqC-stained single-cell suspension (at ~1000 cells/µl, which corresponds to ~20,000 cells) was mixed with the HIV-Seq RT Mix, barcoded using Single Cell VDJ 5' Gel Beads, and partitioned with oil onto a Chromium Next GEM Chip K, following the 10X Genomics protocol for a targeted recovery of 10,000 cells. The chip was then loaded onto a Chromium Controller for single-cell GEM generation and barcoding. RT reactions were performed according to the Chromium Single Cell 5' Reagent Kits v2 (Dual Index) User Guide (10x Genomics CG000330, Rev A). Sequencing libraries were constructed

with 13 cycles of PCR during cDNA amplification and 14 cycles of Sample Index PCR. Gene expression and cell surface protein libraries were pooled and sequenced on a the NovaSeq 6000 lane (S4 flow cell) and sequenced at a minimum of 50,000 reads / cell.

## Data processing, statistics, and analysis

Sequencing libraries from $n = 4$ independent experiments were analyzed. A custom Human/HIV consensus subtype B reference sequence was purpose-built for this study by downloading a Group M alignment from the Los Alamos National Laboratory's HIV sequence database (2018 version), filtering for subtype B sequences, realigning using MAFFT[94] with default settings, and generating a majority consensus sequence using Geneious (http://www.geneious.com/). The consensus HIV sequence was appended to the human reference genome (GRCh38-2020) and annotated as a single exon. Alignment of scRNA-seq reads, collapsing of reads to unique molecular identifier (UMI) counts, and cell calling was performed using CellRanger 6.0.2 (10X Genomics). Filtered count matrices of features generated using the CellRanger count function were then subjected to multiple cleanup steps. Cells with high mitochondrial gene expression (mtDNA% > 15%) and low features (< 500) were removed, along with doublet cells (identified using the DoubletFinder package[95]), in R. All samples were normalized for their transcriptome library depth and batch-corrected between donors using the Seurat[96] NormalizeData and integration function from Seurat package, in R. Harmony batch correction was applied for visualization where indicated. Downstream analysis was performed in SeqGeq (mostly for visualization; FlowJo, LLC) and Seurat v4.3.0. In all analyses, genes are depicted in italics and proteins in bold.

**Clustering.** Graph-based clustering was performed using the Louvain algorithm implementation[97] in the FindClusters Seurat function. The optimal clustering resolution parameters were determined using Random Forests[98] and a silhouette score-based assessment of clustering validity and subject-wise cross-validation[99]. This approach entails evaluating the clustering of cells in a given biological sample based on predictions derived after holding out cells from this sample, and clustering the cells from all the remaining samples, and ensures reproducible clusters across the biological samples. Clusters that appeared biologically under-clustered were subsequently re-evaluated using the same optimal resolution determination procedure and sub-clustered. T cell Receptor Alpha Variable (*TRAV*) and T cell Receptor Beta Variable (*TRBV*) genes were removed from the variable features used for clustering as they were driving the clustering in a donor-specific manner, as expected with randomly-generated VDJ sequences. HIV was also removed from the variable features used for clustering. Thirteen distinct biologically relevant clusters were identified, annotated manually, and used for further analyses.

**Manual gating.** Manual gating was conducted using SeqGeq software to delineate classic CD4 + T cell subsets and to identify HIV-expressing cells. Both gene and protein expressions were used for gating.

**Statistical Analysis of CD4 + T Cell Subsets and HIV RNA+ Cell Distribution.** For establishing associations between CD4 + T-cell subset proportions among the HIV RNA- and HIV RNA+ cells, and HIV RNA+ cell proportions among the different clusters, we used a generalized linear mixed model (GLMM) with a binomial probability distribution implemented in the lme4[100] package in R. In the model, CD4 subset or cluster membership was treated as the outcome being studied. This outcome was represented by comparing the number of cells within the subset/cluster to the number of cells outside of it. The change in subset/cluster membership between HIV RNA status and timing of measurements (viremic vs. suppressed) was estimated as a log odds ratio, defined as the change in the log odds of subset/cluster

membership. This change was estimated with the emmeans[101] R package using the GLMM model fit.

**Differential expressed protein (DEP) analysis.** Raw ADT counts were normalized to 10,000 reads per cell to obtain relative expression values for each marker per cell. Marker expression was then visualized using violin plots, which display data density, median, and interquartile range, to compare the two populations of interest (as indicated in the figure legends) for each donor. Statistical significance was assessed using the Mann–Whitney U test for each marker, followed by False Discovery Rate (FDR) correction of p-values to adjust for multiple comparisons.

**Differential expressed gene (DEG) analysis.** Pseudobulk DEG analysis was performed to identify genes or proteins that were significantly upregulated or downregulated between different cell populations or experimental conditions. DEG analysis at the RNA level was carried out using the R package muscat[102], and counts were summed across clusters and samples. The pbDS function to estimate associations with disease was run with the edgeR[103] method, with minimum cells set to 3 and filter set to gene, and donor was included as a confounder in the model. In the exploratory analysis comparing total CD4 + T cells during viremia vs. suppression, all individuals within a group (viremic or ART-suppressed) were combined, and then gene expression levels were compared across individual cells within each group, followed by implementation of the Wilcoxon rank-sum test to assess for DEGs and proteins, similar to analytical approaches recently described[25,50]. Of note, this analytical approach should be considered exploratory as it does not fully account for correlations between cells from the same subject or paired study designs[104,105]. Results visualized as volcano plots were plotted using EnhancedVolcano[106]. Select genes and proteins of interest among parameters that passed the indicated adjusted p-value (or raw p-value where indicated) thresholds of 0.05 and a $\log_2$ fold-change greater than 0.25 are highlighted in the volcano plots. All genes are depicted in italics and proteins in bold. T cell receptor (TCR) genes (*TRAV* and *TRBV*) were deliberately omitted from the analysis, since private TCR sequences drove donor-specific effects, and the primary focus of our study was to identify gene expression profiles shared across donors.

**Pathway Enrichment Analysis.** The differential gene expression analysis results filtered by adjusted p-value < 0.05 and $\log_2$ fold-change > 0.25 were subjected to over-representation analysis (ORA) using the Enrichr[107–109] web-based tool. Pathway analysis was performed against two curated databases: BioPlanet 2019 and MSigDB Hallmark 2020. This analysis identified significantly enriched pathways and functional categories along with statistical metrics and p-values.

## Distribution of HIV reads

An additional analysis assessing distribution of reads across the HIV genome was performed to determine where HIV transcripts most frequently align. We selected PID8027 and PID1052, for whom we had data on the absence versus presence of HIV-capture primers. Gene expression reads from the week 0 (viremic) timepoint were aligned to a custom, annotated HXB2 reference sequence, which defined discrete HIV coding regions (*5'LTR U3, 5'LTR R, 5'LTR U5, start of gag, gag P1, pol, vif, vpr, rev, vpu, env, tat, nef, 3'LTR U3, 3'LTR R*) using CellRanger (10X Genomics, version 6.0.2). The distribution of HIV reads aligning to these specific regions was then filtered, quantitated using an R script, and compared between samples subjected to the conventional 10X Genomics' 5' scRNA-seq pipeline vs. HIV-Seq.

## Reporting summary

Further information on research design is available in the Nature Portfolio Reporting Summary linked to this article.

## Data availability

Source data are provided with this paper. The sequence datasets generated during this study are available at GEO repository GSE266455 (for PID 0145, 8026, 8027 and 1052; https://www.ncbi.nlm.nih.gov/geo/query/acc.cgi?acc=GSE266455) and at GSE305352 (for datasets generated from people without HIV; https://www.ncbi.nlm.nih.gov/geo/query/acc.cgi?acc=GSE305352). Source data are provided with this paper.

## Code availability

All code used to generate results has been deposited on Github (https://github.com/gladstone-institutes/HIV-seq-_Viremic_vs._ART) and has been archived on Zenodo for reproducibility[110](Gill N., 2025; https://doi.org/10.5281/zenodo.17886108).

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

## Acknowledgements

This work was supported by the National Institutes of Health (P01AI169606 [SAY, NRR, SGD, SAL], R01AI183666 [SAY], R01DK120387 [SAY], R01AI132128 [SAY, JKW], R01AI147777 [NRR], R01DK131526 [NRR], R01AI183286 [NRR], R01AI194343 [ST], R21AI170166 [NRR], UM1AI164559 [NRR], UM1AI164567 [NRR], UM1AI164560 [SGD]) and the California HIV/AIDS Research Program (BB19-SF-009/A135087 [ST]). We also acknowledge support from CFAR (P30AI027763 [ST]) and the James B. Pendleton Foundation. S.T is supported by Doherty Institute for Infection and Immunity Locarnini Fellowship in Virology and University of Melbourne Department of Infectious Diseases Research Support Package. The funders had no role in study design, data collection and analysis, decision to publish, or preparation of the manuscript. We thank Viva Tai and Marian Kerbleski for assistance with the SCOPE specimens; Vivian Pae, Sannidhi Sarvadhavabhatla, and Maria Sophia Donaire for assistance with the Treat Acute HIV specimens; Francoise Chanut for editorial assistance; Robins Givens for administrative assistance; and the study participants for their samples. This work utilized the computational resources of the UCSF Wynton cluster (https://wynton.ucsf.edu)[111].

## Author contributions

J.F and S.T. designed the experiments, performed scRNA-seq experiments, conducted analyses, interpreted data, and prepared figures and tables. X.L. developed pipeline for quality control analysis of scRNA-seq datasets. N.G., R.T., and D.A. performed scRNA-seq analyses, and P.R. created the HIV consensus genome used for scRNA-seq data alignment. R.H. and J.K.W. recruited participants, collected clinical data, and collected biospecimens. S.G.D. oversaw the SCOPE cohort procedures and S.A.L. managed specimen collection. A.J.B., N.R.R., and S.A.Y.

performed supervision. J.F., S.T., N.R.R., and S.A.Y. conceived the study, interpreted data, and wrote the manuscript. N.R.R. and S.A.Y. contributed equally. All authors have read and approved this manuscript.

## Competing interests

The authors declare no competing interests.
