## [Peer Review file · Nature Communications]

HIV-SEQ REVEALS GENE EXPRESSION DIFFERENCES BETWEEN HIV-TRANSCRIBING CELLS FROM VIREMIC AND SUPPRESSED PEOPLE WITH HIV

Corresponding Author: Professor Steven Yukl

Version 0:

Reviewer comments:

Reviewer #1

(Remarks to the Author)

In this paper, Frouard et al. developed a new single-cell sequencing method, HIV-seq, to perform targeted enrichment of HIV transcripts and improve the efficiency for detecting the rare HIV RNA+ cells from PWH. The authors incorporated this method with single-cell CITE-seq and profiled clinical samples from PWH before and after ART treatment. By comparing the phenotypes during viremia and ART, the author described the signatures of HIV-infected cells during viremia and after viral suppression. The major strength of this study is the adding spike-in primers before cells are loaded into the 10x Chromium machine for targeted enrichment increased the capture of HIV RNA+ cells which increased HIV capture rate. The iterative adjustment of spike-in primers is a good endeavor optimizing this assay. The major weakness is the lack of demonstration of sensitivity, specificity, and how these spike-in primers biased cellular transcriptome. The lack of threshold of setting HIV RNA+ reads make the data not trustable, and thus an uninfected donor control is needed to set the appropriate threshold and to avoid false positive calling of HIV+ cells. The cell cluster annotation and differential gene expression analysis are not rigorous and should be improved. Having $p \leq 0.05$, not $p < 0.05$, is a fundamental statistical flaw. If well addressed, this study can bring innovation and value for the field.

Major comments:

1. For any new single-cell assay, a sensitivity and specificity assay, and how these spike-in primers biased cellular transcriptome must be established first. Please primary CD4+ T cells infected with HIV and sorted/enriched (to obtain 100% HIV infection) versus primary CD4+ T cells with no infection, to perform HIV-seq versus regular 10x 5' CITEseq [4 samples: (a) sorted 100% infected CD4 with HIV-seq, (b) sorted 100% infected CD4 with regular 5' CITEseq, (c) uninfected CD4 with HIV-seq, (d) uninfected CD4 with regular 5' CITEseq]. This benchmarking experiment is needed to address: (a) How should the authors set the threshold of HIV detection in uninfected cells, such as using HIV read count or UMI count as a threshold? (b) what's the sensitivity detecting HIV RNA+ cells increase by using HIV-seq?
2. Impact of spike in on transcriptome: In Figure 2A, the author claimed that "there was not transcriptome differences between with and without spike in". This comparison is not rigorous. Importantly, the authors need to perform differential gene expression analysis (shown by Volcano plot) of (a) sorted 100% infected CD4 with HIV-seq versus (b) sorted 100% infected CD4 with regular 5' CITEseq to see whether HIV spike in primers biased the cellular transcriptome. This is because HIV spike in primers may nonspecifically capture human transcriptome, or may eat into the regular cellular transcriptome capture. Starting with the viremic samples proves that the assay "sort of" works but without comparison to the ground truth.
3. Figure 2C and 2D: without an uninfected donor control to set the threshold, the HIV+ cells identified in this plot cannot be distinguished from false positive detection. Again, a negative control and a threshold is needed. For example, "HIV transcript levels ranging from 1 to 1063 HIV reads per cell" – the field has identified that using one HIV RNA read to call HIV RNA+ cell yield false positive detection. Again, uninfected donor HIV-seq and regular CITEseq is needed – otherwise the data of high levels of HIV detection cannot be trusted.
4. Please do not use "adjusted p value ≤ 0.05 " – it has to be < 0.05 , not "less and equal". There are a lot of " $p \leq 0.05$ " in the paper – please correct them all. This is fundamental statistical mistake.
5. Figure 3E. How do you explain that pathway analysis of DEG in Figure 3E did not show cytolytic-related signatures? HIV RNA+ were also enriched in cluster 1 (~30%), what are the signatures for cluster 1?
6. Figure 3F and 3G, Figure 4E, Figure S5. Please do not leave clusters as 1,2,3,4,5,6 without annotation. Please refer to major single cell papers in the immunology, cancer, or HIV field, such as Figure 2A-2B of PMID 31359002 or Figure 3A-3B

of PMID 36536105. All clusters need to have dot plots or signature gene expression and heatmaps (of differentially expressed genes) to justify their cluster name.

7. In Figure 3B, HIV RNA+ cells are enriched in the right part of the cluster which indicates cluster 1 from Figure 3F is a heterogeneous cluster containing at least 2 distinct populations. The resolution for UMAP needs to be improved for further analysis and the inaccurate clustering results in biased conclusion. The same concern is also addressed on Figure 4E and 4F. The authors need to increase the resolution until biologically relevant CD4+ T cell clusters (supported by their cluster annotation by RNA and protein expression, shown by dot plots) are present and reflect human physiology. There are more than 6 different types of CD4s in the peripheral blood, but there are only 6 clusters without biological meanings.
8. Figure 3C, 4C, and 5D showed that HIV+ cells were enriched in Tem. Can you visualize the Tem/Tcm/Tm/Tn/CCR7+, CD37- populations on UMAP? Is the Tem population overlap with HIV RNA+ spots?
9. Figure 4F. To claim HIV enrichment in IL7R and BCL2-high cells, using feature plot in UMAP is not rigorous. The authors should use violin plots of IL7R and BCL2 expression in HIV+ versus HIV- cells and perform statistical test (eg. Wilcox) with Benjamini-Hochberg corrected P values to rigorously justify this statement.
10. Batch effect correction: Seurat default batch effect correction (integration) is known to have problems. For example, Figure S6 shows batch effects not corrected. Please use a different batch effect correction method (such as Harmony) and replot Figure S6 to see if there are still batch effects.
11. Removing HIV reads from clustering: it's great that the authors are aware that "T cell Receptor Alpha Variable (TRAV) and T cell Receptor Beta Variable (TRBV) genes were removed from the variable features used for clustering as they were driving the clustering in a donor-specific manner". In fact, HIV leads to the same bias. Please remove HIV from the clustering process and re-define the clusters.
12. Figure 1 – except for spiking in HIV, Figure 1B is generic from 10x platform that is already commercially available and should not be presented as Figure 1 as if it's a novel design by the authors. Please remove Figure 1B.

Minor comments:

1. Table 1 is not in publication quality. Please change font size, remove grid lines (only horizontal lines)
2. Please describe HIV mapping cut-off (based on uninfected donor results), package, version.
3. Supplement table of differential gene expression – please remove it from Word file and use an Excel file instead so that the following figures are not blocked from reading.
4. Supplement tables: gene names have to be italicized. Figures and text: HIV gene names (gag) should be all lower case and italicized. Please refer to genomics nomenclature guide in Wikipedia or genomics textbooks.
5. Adjusted p-value should be used to analyze DEG and pathway analysis as assessing multiple comparisons on gene expression will increase the chance of false positive cases when using raw p-value.
6. Can the author provide information/data to demonstrate the titration of spike-in HIV probes? What are the differences between spiking in 0.33 μ M vs. 0.67 μ M primers? This will be very helpful for the field.

Reviewer #2

(Remarks to the Author)

Frouard et al. describes an improved method called HIV-seq to identify HIV RNA+ cells using single-cell RNA-seq (scRNA-seq) in samples from people living with HIV. This approach builds upon previous literature to identify HIV RNA+ cells using the 10x Genomics platform. In this manuscript, the authors introduce to the default 10x single cell workflow a capture-based method to target HIV transcripts and enrich cells that are actively transcribing HIV using probes that are specifically targeting B subtype HIV. Currently, detection of HIV RNA+ cells is a major challenge in single-cell studies, particularly at ART when viral transcription is low, thus the authors' methodology could be a significant contribution to aid this area of research. The expectation leading into the results section of the manuscript was that this was largely a methods paper, which follows logically from the main text of the very well-written introduction. However, despite this premise, the results moved very quickly from implementation and validation of the actual method and into a more superficial analysis of differential gene expression with overinterpretation of findings with biology of HIV persistence. Overall, the authors have not provided sufficient evidence to support their claim that this enrichment method increased detection of HIV RNA+ cells over the previous standard 10x method.

Major comments:

Given the small increase in frequency of HIV RNA+ cells, the lack of the total number of cells in the text and figure legends makes accurate interpretation of the HIV RNA+ cells challenging. The absolute numbers of HIV RNA+ and total CD4 T cells per participant with and without HIV-seq will determine if there is actually a substantial increase in cells with HIV and should be included to assess if there is actually a consequential increase in HIV RNA+ cells. This incremental change does not warrant the manuscript title.

On the flip side the authors use absolute numbers to compare HIV RNA without providing the total number of CD4 T cells assessed for each participant. Thus, statements such as in the discussion section page 16 lines 344-352 using absolute numbers are not comparable. For instance, if many, many CD4 T cells from a donor are sequenced, there is higher probability of finding HIV RNA+ cells than when fewer cells are characterized. This impacts even figures like 2D which show comparisons of number of HIV transcripts rather than proportions (are these differences significant). Figures 2A and 3A are missing the total (CD3+CD4+ T cells), as well as number of cells with HIV. Page 23 also does not provide the input cells per 10x well of a chip, and so it is difficult to evaluate the impact of HIV-seq from these experiments.

In the methods section CD4+ T cells were purified using negative selection and then PBMCs from the same donor was spiked back at a ratio of 5:100 in order to have non-CD4+ T cells to establish gates for defining HIV RNA+ cells. Clustering

methods for scRNA-seq such as WNN are well-enough established that such lab approaches are unnecessary. Could this have led to inadvertent contamination with other cell types? There are also a number of Ig genes identified as DEGs in table S3 and S5. Why are immunoglobulin genes expressed in CD4 T cells? Were these variable genes used in the clustering step or is there cell contamination from non-CD4 T cells?

The authors rely heavily on exploratory analyses when they fail to detect significant DEGs between various conditions which emphasizes the problem of small sample sizes. Use of 'nominal p-values isn't necessarily a fatal issue, but here they are repeatedly used, and then downstream analyses are further applied onto these non-significant findings. This overall makes some of the results feel tenuous and less convincing.

The authors generated surface protein data (CITE-seq) – on only a very limited panel of antibodies (15). It's unclear how or where that data is being used. The authors have also performed DEG analyses, but similar DEP or cluster analyses is missing or limited and thus CITE-seq analyses is biased and incomplete.

How was the cutoff for HIV low and high determined? Is this arbitrary, perhaps why a difference is not observed when stratifying by this category (Page 8)? This also drives home the fact we need more HIV RNA+ cells, not just increased reads from the same cells as shown in Fig 2C-D.

There seems to be a marked bias shown in Fig 2D indicating a 3' bias. Can the authors improve this by including probes for the 3' LTR region of the virus? Also given that there are many subtypes of HIV, can the authors expand this method to other non-B subtypes to make this approach more generalizable to other global subtypes?

The authors have validated previous findings shown during viremia that TEMs have the most number of HIV RNA+ cells and express cytolytic markers in a pool of >1000 HIV RNA+ cells. However, their key findings at the ART suppressed timepoint is much weaker as only 26 cells were detected with HIV RNA. Additional samples to increase cell numbers at the ART timepoint like the viremic will be essential to have meaningful results for comparisons and title is thus overstated.

Minor points

Further, as seen in the supplementary tables, a lot of host transcripts are long-noncoding RNA and thus not polyadenylated are sequenced when using the default 10x platform and so authors should tone down their suggestions that only poly-A sequences are targeted.

Page 7 lines 129-130: figure 2A simply shows cluster distributions on UMAPs. This evidence is insufficient to support the assertion of "no difference in global gene expression."

Pages 8,9 lines 174-182: can you show where these subclusters fall on your UMAP? Would go a long way to helping contextualize the data shown in subfigure 3B.

Figure 3F: pie charts are not ideal way to show data. It is very difficult to interpret changes in relative representation for clusters other than 1 and 2.

Are there sufficient cells to do TCR analysis?

Were HIV transcripts included while clustering?

Reviewer #3

(Remarks to the Author)

Version 1:

Reviewer comments:

Reviewer #1

(Remarks to the Author)

It is a pity that instead of addressing concerns and improving rigor, the authors did not respond sufficiently to reviewer suggestions. This manuscript, despite presenting an exciting and innovative spike-in strategy, unfortunately did not demonstrate the rigor compared with other manuscripts at Nature Communications.

Point 1. Sorting pure populations of HIV+ samples is technically feasible (PMID: 30282021). The authors did not attempt to address this point.

Point 3. While this reviewer appreciates the use of uninfected cells as a control as requested, what's the number of cell sequenced, and how is the number comparable with the HIV+ samples? It is showing a lack of rigor.

Point 7. For CD4 T cell resolution: the authors stated to stay with current resolution of 6 clusters without minimal biological annotations. In other existing publications, there are 10 clusters in PMID 35320704, 14 clusters in PMID 36070690, and 15 clusters in PMID 37922905. While it is not the number of cluster that matters - it is whether the authors examined and dissected the biology. See PMID: 33879890, Nature Med - 10 clusters, including naive, central memory, effector memory, Th1, Th2, Th17, Treg, proliferating, in the peripheral blood. Leaving the annotation as cluster numbers without resolution to these essential peripheral blood CD4 cell types does not bring insight to the field. Please at least follow cell type annotations by PMID: 33879890.

Point 12: We requested removal of Figure 1B, which is a commercial 10x Genomics illustration - but Figure 1B remains in this version. The current Figure 1 remains extremely weak. I have never seen any publication with such a weak Figure 1 - probe design, that's it, no data, as a major Figure.

Reviewer #2

(Remarks to the Author)

The authors now provide the frequency of HIV RNA+ cells and the total number of CD4 T cells assessed for each donor and experiment. From these numbers a z-test performed by this reviewer, found no significant differences ($P > 0.05$) for HIV RNA proportions with and without HIV-SEQ for all comparisons. It will be important to justify more precisely than is currently provided, on what basis the inclusion of HIV-1 probes is claimed to increase detection of HIV RNA+ cells. Consequently, although this paper is the second scRNA-seq study with >1000 HIV RNA+ cells detected, the authors have not currently demonstrated that this is because of their method.

Detecting further HIV RNA+ cells at the viremic timepoint has only added incremental insight into HIV RNA+ cells. Not using the ADT data in their differential analysis and generating TCR libraries are missed opportunities. Although the authors sought to perform 'a meaningful analysis of HIV RNA+ cells from ART-suppressed PWH' (line 99), they were only able to detect 25 HIV RNA+ cells at this timepoint in 3 donors which did not allow for a rigorous analysis.

Finally, and most importantly, reluctance to remove HIV while clustering, ineffective batch correction, and the absence of code to replicate their findings makes it challenging to evaluate the findings and suggests a lack of rigor. Only some data generated in the study was submitted to GEO. Improving existing methods is essential, but dependent on sufficient transparency for others to validate results, which is not currently available here.

Reviewer #3

(Remarks to the Author)

Version 2:

Reviewer comments:

Reviewer #1

(Remarks to the Author)

This reviewer recognizes the effort that the authors attempted to revise the manuscript, although the quality of the bioinformatic analysis is not up to the current standard of the field.

These are previously raised points that were not addressed.

Figure 3A and 4E

Please annotate the clusters, instead of 1, 2, 3, 4, 5, 6.

Please integrate uninfected data into Figure 5.

Figure 5A: one UMAP of batch-effect-corrected single cell data including data from viremic, suppressed, and uninfected donors.

Figure 5B and 5D: Add uninfected donor data to the bar graph, as a reference to see how HIV affects these T cells.

Figure 5C: Add differential gene expression volcano plot of viremia vs uninfected, suppressed vs uninfected

Figure 5E: Add uninfected donor data to the bar graph, as a reference to see how HIV affects these T cells.

(Remarks on code availability)

Point-by-point Response to Reviewers

We thank the reviewers for their thoughtful comments. Please find below our point-by-point response to the reviewers' comments *in blue italics*, followed by our responses.

Reviewer #1:

Summary and Major Issues

In this paper, Frouard et al. developed a new single-cell sequencing method, HIV-seq, to perform targeted enrichment of HIV transcripts and improve the efficiency for detecting the rare HIV RNA+ cells from PWH. The authors incorporated this method with single-cell CITE-seq and profiled clinical samples from PWH before and after ART treatment. By comparing the phenotypes during viremia and ART, the author described the signatures of HIV-infected cells during viremia and after viral suppression. The major strength of this study is the adding spike-in primers before cells are loaded into the 10x Chromium machine for targeted enrichment increased the capture of HIV RNA+ cells which increased HIV capture rate. The iterative adjustment of spike-in primers is a good endeavor optimizing this assay.

We thank the reviewer for their thoughtful and constructive feedback. We appreciate their recognition of the strengths of our study, particularly our spike-in of HIV primers to enhance the capture of HIV RNA+ cells.

The major weakness is the lack of demonstration of sensitivity, specificity, and how these spike-in primers biased cellular transcriptome.

As detailed below, we have conducted additional analyses to further demonstrate the sensitivity and specificity of our approach, and to confirm that the spike-in primers do not bias the cellular transcriptome.

The lack of threshold of setting HIV RNA+ reads make the data not trustable, and thus an uninfected donor control is needed to set the appropriate threshold and to avoid false positive calling of HIV+ cells.

As detailed below, we have addressed this concern by performing new experiments using uninfected donor controls. These experiments allowed us to establish an appropriate threshold and minimize the risk of false-positive classification of HIV+ cells.

The cell cluster annotation and differential gene expression analysis are not rigorous and should be improved.

As detailed below, we have now provided additional information and clarification on the clustering and differential gene expression analysis approaches we had used, which had implemented rigorous statistical methodologies.

Having $p \leq 0.05$, not $p < 0.05$, is a fundamental statistical flaw.

We have now clarified that all our " $p \leq 0.05$ " statements were in fact $p < 0.05$, and we changed the manuscript accordingly.

If well addressed, this study can bring innovation and value for the field.

Major comments:

1. For any new single-cell assay, a sensitivity and specificity assay, and how these spike-in primers biased cellular transcriptome must be established first. Please primary CD4+ T cells infected with HIV and sorted/enriched (to obtain 100% HIV infection) versus primary CD4+ T cells

with no infection, to perform HIV-seq versus regular 10x 5' CITEseq [4 samples: (a) sorted 100% infected CD4 with HIV-seq, (b) sorted 100% infected CD4 with regular 5' CITEseq, (c) uninfected CD4 with HIV-seq, (d) uninfected CD4 with regular 5' CITEseq]. This benchmarking experiment is needed to address: (a) How should the authors set the threshold of HIV detection in uninfected cells, such as using HIV read count or UMI count as a threshold? (b) what's the sensitivity detecting HIV RNA+ cells increase by using HIV-seq?

We agree that establishing the sensitivity and specificity of any new single-cell assay is essential, as well as assessing how spike-in primers may bias the cellular transcriptome. However, we disagree that the specific experimental conditions requested by the reviewer are necessary to achieve these goals. In particular, the suggested experiment would not work for multiple reasons. First, it is not possible to obtain a population of 100% infected primary CD4+ T cells, even using a VSV-G pseudo typed virus, which circumvents the requirement for CD4 and co-receptor (CCR5/CXCR4) engagement¹. Even if one were to use an X4-tropic reporter HIV virus and to use *ex vivo* stimulated CD4+ T cells to increase cellular permissivity, at any given time only a few percent of the cells would be infected. Based on extensive experience in working with such infected primary cells, bead-based enrichment would only marginally increase the proportions of infected cells, while sorting of HIV-infected primary cells would be harsh on the infected cells (which are more fragile than uninfected cells) and result in cells of insufficient viability and quantity for follow-up assays. Instead, the only way to obtain a pure population of 100% infected cells would be to use a cell line model – e.g., one of the JLAT clones, as compared to uninfected Jurkat cells. However, all JLAT cells – even amongst a single clone – exhibit a range of HIV RNA expression levels, as we have previously demonstrated at the single cell level². Even sorting of GFP+ JLAT cells will result in a population of cells that will re-distribute into GFP+ and GFP- cells. This would pose a problem in a benchmarking HIV-seq analysis, since one cannot tell whether lack of detection of HIV RNA in that cell is due to a sensitivity issue, or simply because that cell is not expressing HIV RNA. Furthermore, JLATs are very different from primary CD4+ T cells from patients, which is what HIV-seq is designed for.

We fully agree, however, that it is important to establish a clear threshold of HIV detection in uninfected cells. Hence, as detailed below in our response to comment #3, we have addressed this question by performing HIV-seq on both cells from PWH as well as cells from people without HIV (PWOH), and we demonstrated that no HIV+ cells were detected in the cells from the PWOH. These new data are presented in the revised manuscript (Lines 137-141 and in Fig.S3).

To address the increase in sensitivity of detection HIV RNA+ cells by using HIV-seq, we had directly compared CD4+ T cells from PWH processed using HIV-seq versus regular 5' CITE-seq (Lines 132-139 and in Fig. 2B and C in original manuscript). These results had showed that HIV-seq significantly improved the sensitivity of detecting of HIV+ cells, and in the revised manuscript are presented in Lines 142-150 and Fig. 2B and C.

The issue of establishing how the spike-in primers may affect the cellular transcriptome is addressed in point #2 immediately below.

2. Impact of spike in on transcriptome: In Figure 2A, the author claimed that “there was not transcriptome differences between with and without spike in”. This comparison is not rigorous. Importantly, the authors need to perform differential gene expression analysis (shown by Volcano plot) of (a) sorted 100% infected CD4 with HIV-seq versus (b) sorted 100% infected CD4 with regular 5' CITEseq to see whether HIV spike in primers biased the cellular transcriptome. This is because HIV spike in primers may nonspecifically capture human transcriptome, or may eat into the regular cellular transcriptome capture. Starting with the viremic samples proves that the assay “sort of” works but without comparison to the ground truth.

As detailed above in Response #1, it is not possible to obtain a 100% pure population of cells where every cell is known to harbor HIV RNA. However, such a population is not needed to address this point the reviewer makes, that it is very important to formally demonstrate that HIV-seq does not bias the cellular transcriptome. Although the UMAPs in Fig. 2A revealed no obvious change in cellular transcriptome upon addition of the HIV capture sequences, we agree with the reviewer that a more quantitative assessment is warranted to determine whether any transcriptional changes are elicited by HIV-seq. Hence, we performed differential gene expression (DEG) analysis on the two patient specimens that were analyzed in the absence vs. presence of HIV-seq. These new results are now presented as volcano plots (Fig. S2 in the revised manuscript) and in new supplementary tables (Tables S3 and S4), and we have included them here.

In total, implementation of HIV-seq resulted in only 4 DEGs in PID1052 and 6 in PID8027. The only gene common to both, *AL138963.4*, is a long non-coding RNA³ with limited characterization. Given that such low numbers of DEGs are in line with what would be expected with technical replicates of scRNA-seq analysis of the same sample, we believe our original conclusion that there is no significant global gene expression difference between the two experimental conditions holds. These additional data are now presented in Lines 129-133 of the revised manuscript.

3. Figure 2C and 2D: without an uninfected donor control to set the threshold, the HIV+ cells identified in this plot cannot be distinguished from false positive detection. Again, a negative control and a threshold is needed. For example, “HIV transcript levels ranging from 1 to 1063 HIV reads per cell” – the field has identified that using one HIV RNA read to call HIV RNA+ cell yield false positive detection. Again, uninfected donor HIV-seq and regular CITEseq is needed – otherwise the data of high levels of HIV detection cannot be trusted.

We thank the reviewer for this important suggestion, and we agree that having uninfected donors as a control to set the threshold of HIV+ cells is a better control than our original negative control that consisted of comparing reads to those observed in CD8+ T cells, which are not permissive to HIV. Hence, we have now run side-by-side HIV-seq vs regular 5' scRNA-seq on two uninfected donors. These results are presented here:

It demonstrates that among all of the uninfected samples, not a single HIV transcript was identified, either in the absence or presence of HIV-seq, in contrast to our infected donor PID8027, where HIV RNA+ cells were detected using both methods. Therefore, we are confident that our threshold of 1 HIV read is appropriate for defining HIV+ cells. These new data, which have been added to the manuscript as Fig.S3B, are now discussed in Lines 137-141 in the Results of the manuscript.

4. Please do not use “adjusted p value ≤ 0.05” – it has to be <0.05, not “less and equal”. There are a lot of “p ≤ 0.05” in the paper – please correct them all. This is fundamental statistical mistake.

We have clarified that all the p-values we had indicated were in fact $p < 0.05$, not $p \leq 0.05$. This correction has been made in all figures and tables.

5. Figure 3E. How do you explain that pathway analysis of DEG in Figure 3E did not show cytolytic-related signatures?

We would like to clarify that the conclusion about enrichment of HIV RNA+ cells among cytolytic CD4+ T cells was made in Fig. 3F and G, where we demonstrated that the majority of HIV RNA+ cells belonged to cluster 2, which harbored a cytolytic signature. By contrast, Fig. 3E (and 3D) corresponded to a different type of analysis, where all HIV RNA+ cells were analyzed as a single population of cells and compared to all HIV RNA- cells. Since HIV RNA+ cells are in clusters besides cluster 2, and since cluster 2 includes both HIV RNA+ and HIV RNA- cells, it is not unexpected that the cytolytic signature did not come out of the total DEG analysis (Fig. 3D) nor the pathway analysis of those DEGs (Fig 3E). In essence, the difference relates to the fact that Fig. 3D and E consider HIV RNA+ cells as a uniform bulk population of cells, in contrast to Fig. 3F and G, which leverage the heterogeneity of the HIV RNA+ cells.

HIV RNA+ were also enriched in cluster 1 (~30%), what are the signatures for cluster 1?

Cluster 1 cells expressed high levels of SELL, TCF7, IL7R, RACK1, LEF1 and CCR7, suggesting their identity to be a central memory (Tcm) cells and naïve cells. The unique features of this cluster, as well as the other clusters analyzed, are now presented as a new supplementary figure, Fig. S8. The persistence of HIV in this cluster is consistent with known persistence of HIV in the Tcm compartment⁴, which is now discussed in the paper (Lines 268-270).

6. Figure 3F and 3G, Figure 4E, Figure S5. Please do not leave clusters as 1,2,3,4,5,6 without annotation. Please refer to major single cell papers in the immunology, cancer, or HIV field, such as Figure 2A-2B of PMID 31359002 or Figure 3A-3B of PMID 36536105. All clusters need to have dot plots or signature gene expression and heatmaps (of differentially expressed genes) to justify their cluster name.

We have now annotated the six clusters as requested. Clusters were annotated based on DEGs expressed in each of the clusters (using the FindClusters function in Seurat), as well as by

manually assessing for expression levels of markers of classic cellular subsets. In summary, cluster 1 was annotated as a mix of central memory and naïve CD4+ T cells (defined by high expression of *SELL*, *TCF7*, *IL7R*, *RACK1*, *LEF1*, *CCR7*); clusters 2 and 5 as two types of effector/cytolytic CD4+ T cells (defined by their expression of *GNLY*, *PRF1*, *GZMA*, *GZMB*, *GZMH*, *GZMM*, *LAMP1*), with cluster 2 expressing higher levels of all cytotoxic markers as compared to cluster 5; cluster 3 as proliferating CD4+ T cells (defined by high expression of *MKI67*, *CDK1*, *PCNA*, *MCM2*, *TYMS*, *RRM2*, *TOP2A*, *RACGAP1*); cluster 4 as a small cluster of platelet-CD4+ T cell aggregates previously described in scRNA-seq datasets⁵ (and defined by high expression of *GP9*, *PF4*, *PPBP*); and cluster 6 as activated HLA-DR-expressing CD4 + T cells (defined by high expression of *HLA-DRB1*, *HLA-DPA1*, *HLA-DPB1*, *CD74*). We have now updated Figures 3F and 4E and the manuscript text accordingly (Lines 228-234) with the annotations. We have also, as requested, displayed the signature gene expression profiles of each of the clusters in dot plot format, to justify the cluster names (new Fig. S8). The new Fig. S8 is presented here:

7. In Figure 3B, HIV RNA+ cells are enriched in the right part of the cluster which indicates cluster 1 from Figure 3F is a heterogeneous cluster containing at least 2 distinct populations. The resolution for UMAP needs to be improved for further analysis and the inaccurate clustering results in biased conclusion.

Although clustering can be performed in a subjective manner to increase the numbers of clusters using user-defined criteria, the current standard in the scRNA-seq field is to use objective criteria to prevent over- or under-clustering⁶. Otherwise, with user-selected choice of the clustering resolution parameter, one could derive as many or as few clusters as desired without solid biological basis. We opted against user-selected choice of clustering since it is currently advised against by statisticians in the scRNA-seq field, as it results in different types of clustering between different scRNA-seq studies. Instead, we implemented an approach that is now common in scRNA-seq analysis (and detailed in our Methods section Lines 576-578 of the initial manuscript, Lines 619-621 of the revised manuscript), where the criteria for choosing the correct resolution is based objectively and statistically as one that resulted in reproducible clusters across biological samples in the study. More specifically, by using a cross-validation approach, we evaluated the clustering of cells in a given biological sample based on predictions derived after holding out cells

from this sample, and clustering the cells from all the remaining samples, as described in recent studies^{7,8}. We have now clarified in our revised manuscript that our clustering method uses a cross-validation approach to objectively establish the numbers of clusters in our dataset (Lines 621-624).

The same concern is also addressed on Figure 4E and 4F. The authors need to increase the resolution until biologically relevant CD4+ T cell clusters (supported by their cluster annotation by RNA and protein expression, shown by dot plots) are present and reflect human physiology. There are more than 6 different types of CD4s in the peripheral blood, but there are only 6 clusters without biological meanings.

With regards to the numbers of types of CD4+ T cells in peripheral blood, or in fact the number of types of any subset of cells in any compartment, such a precise number cannot be established, as it depends on what specific pathway or readout one is looking at. Indeed, one could consider there to be millions of types of CD4+ T cells in blood, as every individual CD4+ T cell has a different transcriptional signature. We would also like to caution against making conclusions by looking at UMAP and shapes of clusters in the UMAP. As discussed by Chari et al., generating UMAP figures for exploratory cell subset annotation can generate false clusters⁹. Although user-determined cluster resolution can be useful for analyses where one wants to make a point about specific aspects of biology, we disagree with the suggestion to manually increase cluster resolution, and instead opt for the standard, objective clustering that is based on statistical rather than subjective measures, as detailed above and now in Lines 619-624.

8. Figure 3C, 4C, and 5D showed that HIV+ cells were enriched in Tem. Can you visualize the Tem/Tcm/Tm/Tn/CCR7+, CD37- populations on UMAP?

We have now added a visualization of the classical CD4+ T-cell subsets on the UMAP in the new Fig. S7A, and this is presented here:

Is the Tem population overlap with HIV RNA+ spots?

By comparing Fig. 3F with the new Fig. S7A, one can see that cluster 2, where HIV RNA+ cells are enriched, also exhibits the most overlap with Tem cells. This is assessed more quantitatively in the new Fig. S7B (and presented below), where we show using pie graphs that among all the clusters, cluster 2 is the most enriched for Tem cells. Hence, the Tem population does indeed overlap with the location of the HIV RNA+ cells.

9. Figure 4F. To claim HIV enrichment in IL7R and BCL2-high cells, using feature plot in UMAP is not rigorous. The authors should use violin plots of IL7R and BCL2 expression in HIV+ versus HIV- cells and perform statistical test (eg. Wilcox) with Benjamini-Hochberg corrected P values to rigorously justify this statement.

In response to these comments, we first performed a statistical analysis of IL7R expression in HIV+ versus HIV- cells using Wilcoxon rank-sum tests with Benjamini-Hochberg correction:

We found that the data were more effectively visualized as bar graphs (top) rather than as violin plots (bottom), because of the relatively high numbers of HIV- cells forming the “tails”, making it visually look like expression levels were higher among the HIV- cells when in fact it is in fact higher among the HIV+ cells. Hence, we have opted to display the comparison as bar graphs, which are now presented as the new Fig. 4F. These results confirmed our initial report that IL7R expression is higher among HIV+ vs HIV- cells.

When we performed a similar analysis for BCL2 expression, we found that the difference in BCL2 expression was not statistically different between HIV+ and HIV- cells:

We noted, however, that among all the HIV RNA+ cells, 44% expressed BCL2. Notably, all of these BCL2+ HIV RNA+ cells expressed IL7R, and among the IL7R+ HIV RNA+ cells, 52.4% expressed BCL2:

Overall, these results that IL7R and BCL2 are commonly co-expressed, and that a substantial proportion of HIV RNA+ reservoir cells express both IL7R and BCL2, which was the original point we had set out to make. These new analyses are now updated in the Figures 4F and G and discussed in Lines 270-275 of the revised manuscript.

10. Batch effect correction: Seurat default batch effect correction (integration) is known to have problems. For example, Figure S6 shows batch effects not corrected.

We would first like to point out that Seurat integration has been shown to be among the top three batch correction methods in a widely cited benchmark study¹⁰. We had opted for Seurat integration in our original analysis because it returns a batch-corrected count matrix that is then used for differential gene expression analysis. Harmony, another top-ranked batch correction method¹⁰, corrects low-dimensional representations (e.g., PCA, UMAP), but does not return a batch-corrected count matrix, which complicates downstream analyses such as differential gene expression analysis.

Please use a different batch effect correction method (such as Harmony) and replot Figure S6 to see if there are still batch effects.

In response to the reviewer's suggestion, we applied Harmony as an alternative batch correction method and re-plotted the original analyses performed in Figure S6. The analysis revealed that the batch effect persisted both before (top row) after (bottom row) correction by Harmony:

Given this result and the fact that Harmony does not return a batch-corrected count matrix, we have opted to maintain our original approach and have not included the Harmony-corrected data in the revised manuscript. However, we would be happy to include the plots above as a supplementary figure if the reviewer feels this is important.

11. Removing HIV reads from clustering: it's great that the authors are aware that "T cell Receptor Alpha Variable (TRAV) and T cell Receptor Beta Variable (TRBV) genes were removed from the variable features used for clustering as they were driving the clustering in a donor-specific manner". In fact, HIV leads to the same bias. Please remove HIV from the clustering process and re-define the clusters.

We would first like to clarify that our rationale for removing TCR genes from the clustering analysis was to avoid donor-specific clustering, as TRAV and TRBV genes have donor-specific features. This donor-specific clustering would not be expected for HIV RNA transcripts, which were detected in all the donors (i.e., unlike TRAV/TRBV, they were not unique to a particular donor).

An issue separate from the donor-specific clusters is whether HIV transcripts drove the clustering overall (as opposed to donor-specific clusters). Multiple analyses suggested that they did not. First, using Receiver Operating Characteristic (ROC) analysis¹¹, we found that HIV transcripts were not among the top 20 genes driving the clustering. These data are now included as a table entitled "Top20genes_clusters" that can be found after the references listed at the end of this document. Second, we removed HIV transcripts and re-clustered the cells using the same principal components (PCs) and resolution. We then compared cluster membership before vs. after HIV transcript removal. Visualization of the data by UMAPs demonstrated that overall cluster distribution was similar with (left) vs. without (right) HIV transcript inclusion:

Third, for a more quantitative analysis, we examined the confusion matrix comparing cluster membership before vs. after HIV transcript removal. This approach enables assessment of the extent to which cells moved between clusters before vs. after HIV transcript removal.

In the confusion matrix plot above, the boxes show what proportion of cells remained in the same cluster before and after removing HIV transcripts. The brighter the color, the more cells stayed put. The observation that the yellow/green boxes are all along the diagonal (corresponding to the condition where each cluster identity stayed put), while all other boxes are purple, demonstrated that cluster membership overall did not overall change after HIV transcript removal. Finally, to formally quantitate cluster stability, we calculated the adjusted Rand index (ARI) score, a reflection of how similar two types of clustering are. An ARI score of 1 corresponds to 100% identity. We found that the ARI score comparing clustering with vs. without HIV transcripts was very close to 1 (0.96), demonstrating that removing HIV transcripts had minimal effect on clustering. Based on all these results, we have opted to keep the HIV transcripts in our clustering,

as the most unbiased method of clustering and also because it did not markedly change cluster distribution. We have not included in the revised manuscript the above analyses, but would be happy to do so if the reviewer feels it is important.

12. Figure 1 – except for spiking in HIV, Figure 1B is generic from 10x platform that is already commercially available and should not be presented as Figure 1 as if it's a novel design by the authors. Please remove Figure 1B.

In response to this comment, we have now removed the portion of Figure 1 that referred to the commercially available 10X platform pipeline while retaining the general workflow of HIV-seq.

Minor comments:

1. Table 1 is not in publication quality. Please change font size, remove grid lines (only horizontal lines)

The formatting issue has been addressed in the revised manuscript.

2. Please describe HIV mapping cut-off (based on uninfected donor results), package, version.

As detailed above in the response to question #3, to establish the threshold for detecting HIV-positive cells and transcripts, we have now performed HIV-seq vs. the regular scRNA-seq pipeline on samples from uninfected donors (new Fig S3B and Lines 137-141) to establish that that our threshold of 1 HIV read is appropriate to define HIV+ cells, since no HIV transcripts were identified in any of the uninfected samples, irrespective of HIV-seq. Our mapping was done using CellRanger software version 6.0.2 (from 10X Genomics), as indicated in the original manuscript Line 565 (now Line 608 in revised manuscript).

3. Supplement table of differential gene expression – please remove it from Word file and use an Excel file instead so that the following figures are not blocked from reading.

The supplementary tables of differential gene expression are now all presented in the excel file entitled “Supplementary_Tables”, as requested.

4. Supplement tables: gene names have to be italicized. Figures and text: HIV gene names (gag) should be all lower case and italicized. Please refer to genomics nomenclature guide in Wikipedia or genomics textbooks.

We have updated the supplementary tables, figures, and text to ensure gene names follow the correct formatting guidelines.

5. Adjusted p-value should be used to analyze DEG and pathway analysis as assessing multiple comparisons on gene expression will increase the chance of false positive cases when using raw p-value.

Throughout the manuscript, we primarily reported adjusted p-values – these p-values were implemented in Fig S3D, Fig. 3D + Table S3, Fig. 4D + Table S4, and Fig. S7 + Table S6 of the original manuscript (now Fig S5D, Fig. 3D + Table S5, Fig. 4D + Table S6, and Fig. S10 + Table S8 in the revised manuscript). Unadjusted p-values were only used for two analyses that did not yield statistically significant DEGs after multiple testing correction - Fig. 5C + Table S5, and Fig. 5E + Table S7 of the original manuscript (now Fig. 5C + Table S7, and Fig. 5E + Table S9 in the revised manuscript), which were considered exploratory analyses. We had clearly indicated the exploratory nature of these two analyses by stating: “As an exploratory analysis, however, we assessed the DEGs with the lowest raw p-values to gain insights into potential cellular pathways” Lines 265-267 (now Lines 288-291 in the revised manuscript) and “As an exploratory analysis, we assessed the identities of the DEGs based on the raw p-values” Lines 299-300 (now Lines 322-323 in the revised manuscript).

6. Can the author provide information/data to demonstrate the titration of spike-in HIV probes? What are the differences between spiking in 0.33 μM vs. 0.67 μM primers? This will be very helpful for the field.

In order to determine the optimal concentration for the spike-in HIV primers, we had tested different concentrations across 3 experiments and on cells from both viremic and suppressed samples. These raw data were not presented in the original manuscript, but given this request we now include them in the new Fig. S1. In experiment 1, we had first compared the effect of adding 0.67 μM primers versus no primers on the cDNA recovery yield. We observed a significant reduction in cDNA yield, suggesting that 0.67 μM was too high of a concentration. This finding was true for both suppressed and viremic samples. Hence, we next reduced the amount of primer by half in two other experiments (to 0.33 μM) and found that, compared to no primers, this lower condition did not impair cDNA yield. The cumulative results of all three experiments, which led us to choose the 0.33 μM concentration, are presented in the new Fig. S1 and are depicted below:

Reviewer #2

Frouard et al. describes an improved method called HIV-seq to identify HIV RNA+ cells using single-cell RNA-seq (scRNA-seq) in samples from people living with HIV. This approach builds upon previous literature to identify HIV RNA+ cells using the 10x Genomics platform. In this manuscript, the authors introduce to the default 10x single cell workflow a capture-based method to target HIV transcripts and enrich cells that are actively transcribing HIV using probes that are specifically targeting B subtype HIV. Currently, detection of HIV RNA+ cells is a major challenge in single-cell studies, particularly at ART when viral transcription is low, thus the authors' methodology could be a significant contribution to aid this area of research. The expectation leading into the results section of the manuscript was that this was largely a methods paper, which follows logically from the main text of the very well-written introduction.

While our study introduces HIV-seq as an improved method for detecting HIV RNA+ cells in scRNA-seq, we do not view it as a methods paper, as it provides the first extensive scRNA-seq characterization of HIV RNA+ cells from ART-suppressed people with HIV (PWH) and their direct comparison to HIV RNA+ cells during viremia from the same individuals. Such comparisons could not previously be performed due to insufficient numbers of HIV RNA+ cells that could be recovered by scRNA-seq analysis of specimens from ART-suppressed PWH. We also note that the characterization of 1,072 HIV RNA+ from viremic individuals is the largest such dataset to date, which will be a useful resource for the field.

However, despite this premise, the results moved very quickly from implementation and validation of the actual method and into a more superficial analysis of differential gene expression with overinterpretation of findings with biology of HIV persistence.

We are not clear exactly how the reviewer finds our differential gene expression analysis “superficial” nor which aspect of our findings he/she felt we were overinterpreting, but we would like to point out that our gene expression analyses were all performed using rigorous statistical methodologies. We used a stringent QC pipeline to ensure analysis of only high-quality single-cell data (e.g., removing cells with high mitochondrial gene expression (>15% mtDNA) and low feature counts (<500)), and additionally removed doublets using the recently published approach DoubletFinder¹². Random Forests and silhouette score-based assessment of clustering validity and subject-wise cross-validation approach were implemented to ensure a statistically-validated method of clustering. For gene expression analysis, we aggregated cell-level data into pseudobulk samples grouped by donor before applying edgeR^{13,14} (a differential expression testing method for RNA-seq datasets) which improves the power of differential expression analysis by accounting for inter-individual variability, thereby reducing false positives¹³. Finally, we had reported p-values that passed the 0.05 thresholds after adjustment for multiple correction, and log₂ fold-change > 0.25, as used by others in the field^{15–17}; in two instances only – where we clearly stated that we were performing an exploratory analysis – we reported results that were found based on adjusted p-values, as detailed further below in the response to Comment #4. If there are specific aspects of differential gene expression or specific instances of overinterpretation of findings that were not addressed in the comments that follow, we would be happy to address them.

Overall, the authors have not provided sufficient evidence to support their claim that this enrichment method increased detection of HIV RNA+ cells over the previous standard 10x method.

As detailed further in the responses to Comments #1 and #2 below, we had shown that HIV-seq increases both the number of HIV RNA+ cells detected (by up to 44%) (Fig.2B/Lines 134-137 in original, Fig.2B/Lines 142-148 in revision) and the number of HIV transcripts per cell (with the mean reads per cell increasing from 22 to 44) (Fig.2C/Lines 137-139 in original, Fig.2C/Lines 148-150 in revision).

Major comments:

1. Given the small increase in frequency of HIV RNA+ cells, the lack of the total number of cells in the text and figure legends makes accurate interpretation of the HIV RNA+ cells challenging.

The absolute numbers of HIV RNA+ and total CD4 + T cells per participant with and without HIV-seq will determine if there is actually a substantial increase in cells with HIV and should be included to assess if there is actually a consequential increase in HIV RNA+ cells. This incremental change does not warrant the manuscript title.

We had initially only reported the frequencies of HIV RNA+ cells with versus without HIV-seq, and in response to this request have now additionally added the absolute numbers of both HIV RNA+ cells and total CD4 + T cells (Lines 142-148 in revised manuscript). We note that the manuscript title does not say anything about the magnitude of increase in detection of HIV RNA+ cells. Instead, it focuses on the notion that the host gene expression profiles are different among HIV RNA+ cells during viremia vs. during ART suppression. Our data strongly support this finding, irrespective of the magnitude of increase that HIV-seq enables in recovery of HIV RNA+ cells.

2. On the flip side the authors use absolute numbers to compare HIV RNA without providing the total number of CD4 + T cells assessed for each participant.

We have now added the total number of CD4 + T cells assayed in our study (Lines 166 and 256) and per participant (Figure legend Lines 994-996 and 1028-1030). Specifically, 85,667 CD4+ T cells from 4 viremic samples (13,035, 35,849, 17,723, and 19,060 CD4+ T cells for PID1052, PID8026, PID8027, and PID145, respectively) and 75,158 CD4+ T cells from 3 suppressed samples (15,468, 42,618 and 17,072 CD4+ T cells for PID1052, PID8026 and PID8027, respectively) were analyzed.

Thus, statements such as in the discussion section page 16 lines 344-352 using absolute numbers are not comparable. For instance, if many, many CD4 + T cells from a donor are sequenced, there is higher probability of finding HIV RNA+ cells than when fewer cells are characterized.

As detailed above, our study had sequenced a total of 85,667 CD4+ T cells to identify 1,072 HIV RNA+ cells during viremia. We have now detailed in the aforementioned Discussion section how our having identified many more HIV RNA+ cells was not due to our having sequenced many more cells than prior studies. Specifically, the Collora et al.¹⁵ study, which had identified 81 HIV RNA+ cells, had come from sequencing 52,473 CD4+ T cells. Comparing to our study, we had sequenced 1.6x more cells, but found 13.2x more HIV RNA+ cells, demonstrating that our finding more HIV RNA+ cells was not because we had sequenced many more cells. The other two prior studies that had used single-cell sequencing had identified 164 HIV RNA+ cells from sequencing 95,872 PBMCs (in Geretz et al.¹⁷), and 256 HIV RNA+ cells from sequencing 25,778 memory CD4+ T cells (Wei et al.¹⁶). Because the exact numbers of CD4+ T cells these corresponded to were not provided in these manuscripts, such a direct comparison as performed with the Collora et al.¹⁵ study is not possible. Regardless, if we approximate that CD4+ T cells comprise up to 60% of PBMCs¹⁸, then we estimate that 57,523 CD4+ T cells were sequenced in the Geretz et al.¹⁷ study, which equates to our having sequenced 1.5x more CD4+ T cells but having identified 6.5x more HIV RNA+ cells. Likewise, for the Wei et al.¹⁶ study, if we assume that total CD4+ T cells in PBMCs are comprised of ~50% of memory CD4+ T cells¹⁹, then that study sequenced the equivalent of 51,556 total CD4+ T cells. This information equates to our having sequenced 1.6x more CD4+ T cells but detected 4.2x more HIV RNA+ cells. Hence, these results demonstrate that our recovery of HIV RNA+ cells was unlikely the result of sequencing more input cells. We have now modified the manuscript to better reflect this point (Lines 375-385).

This impacts even figures like 2D which show comparisons of number of HIV transcripts rather than proportions (are these differences significant).

In response to this comment, we have modified Fig. 2D to report HIV transcripts as a proportion of CD4 + T cells rather than as absolute numbers. As seen below, the original data reporting absolute numbers of reads (left) look very similar to the updated figure showing normalized data reporting their proportions (right):

Figures 2A and 3A are missing the total (CD3+CD4+ T cells), as well as number of cells with HIV. We believe the reviewer was referring to Figures 3A and 4A since Figure 2A is a UMAP. We have now added the total number of CD4+ T cells analyzed to these figure panels (Line 167 for Fig. 3A and Line 256 for Fig. 4A). For consistency, in other figure panels where in the original manuscript we had reported the absolute numbers and/or frequencies of HIV RNA+ cells, we now also always include the absolute numbers of CD4+ T cells as well (Lines 994-996 and 1028-1030 in the revised manuscript).

Page 23 also does not provide the input cells per 10x well of a chip, and so it is difficult to evaluate the impact of HIV-seq from these experiments.

We had detailed in the original manuscript that we had added 20 µl of cells at ~1,000 cells/µl into the reaction (Lines 545-546 in original manuscript). This amount corresponds to ~20,000 input cells when one does the math. In the revised manuscript, we have now explicitly stated that we had inputted ~20,000 cells (Lines 588).

3. In the methods section CD4+ T cells were purified using negative selection and then PBMCs from the same donor was spiked back at a ratio of 5:100 in order to have non-CD4+ T cells to establish gates for defining HIV RNA+ cells. Clustering methods for scRNA-seq such as WNN are well-enough established that such lab approaches are unnecessary. Could this have led to inadvertent contamination with other cell types?

The PBMC spike-in was included in order to provide a population of cells non-permissive to HIV infection (e.g., CD8+ T cells) to serve as negative controls for HIV RNA+ cell identification. This population was only used in the initial analysis shown in the original Figure S1 (now Fig. S3). For all downstream analyses, we focused on a rigorously gated population of CD4+ T cells, defined as CD19-/CD20-, CD45+, CD3+, and CD8- cells. Furthermore, all samples were subjected to doublet cell cleanup step using DoubletFinder¹², minimizing the likelihood of doublets between CD4+ T cells and other cell types. Given these stringent QC and upstream gating criteria, contamination with other PBMC-derived cell types would be minimal.

There are also a number of Ig genes identified as DEGs in table S3 and S5. Why are immunoglobulin genes expressed in CD4+ T cells?

Although we had followed 10X Genomics guidelines for cell loading to keep the doublet rate low (by loading a maximum number of ~20,000 cells to keep the doublet rate below ~7.6%), and implemented DoubletFinder as an additional method to bioinformatically exclude doublets, these approaches are not able to remove every last doublet. This result is expected and unavoidable in scRNA-seq experiments. We suspect that the presence of Ig genes in our DEG tables is due to some residual doublets between CD4+ T cells and B cells. A second possibility is that sample processing led to some lysis of B cells, which released B cell-derived ambient RNA which got encapsulated with CD4+ T cells. Both scenarios are well-known technical limitations of single-cell RNA-seq platforms.

Were these variable genes used in the clustering step or is there cell contamination from non-CD4+ T cells?

Since we could not determine the exact source of Ig gene signals in our data, and because there could be other non-Ig genes that resulted from similar kinds of contamination, to be unbiased as possible, we opted to leave all genes that passed QC in during the clustering steps, rather than selectively removing ones that did not “make sense”.

4. The authors rely heavily on exploratory analyses when they fail to detect significant DEGs between various conditions which emphasizes the problem of small sample sizes. Use of ‘nominal p-values isn’t necessarily a fatal issue, but here they are repeatedly used, and then downstream analyses are further applied onto these non-significant findings. This overall makes some of the results feel tenuous and less convincing.

We would like to point out that throughout the manuscript, we primarily reported adjusted p-values – these p-values were implemented in Fig S3D, Fig. 3D + Table S3, Fig. 4D + Table S4, and Fig. S7 + Table S6 of the original manuscript (now Fig S5D, Fig. 3D + Table S5, Fig. 4D + Table S6, and Fig. S10 + Table S8 in the revised manuscript). Unadjusted p-values were only used for two analyses that did not yield statistically significant DEGs after multiple testing correction - Fig. 5C + Table S5, and Fig. 5E + Table S7 of the original manuscript (now Fig. 5C + Table S7, and Fig. 5E + Table S9 in the revised manuscript), which were considered exploratory analyses. We had clearly indicated the exploratory nature of these two analyses by stating: “As an exploratory analysis, however, we assessed the DEGs with the lowest raw p-values to gain insights into potential cellular pathways” Lines 265-267 (now Lines 288-290 in the revised manuscript) and “As an exploratory analysis, we assessed the identities of the DEGs based on the raw p-values” Lines 299-300 (now Lines 322-323 in the revised manuscript).

5. The authors generated surface protein data (CITE-seq) – on only a very limited panel of antibodies (15). It’s unclear how or where that data is being used. The authors have also performed DEG analyses, but similar DEP or cluster analyses is missing or limited and thus CITE-seq analyses is biased and incomplete.

Our CITE-seq panel was intentionally designed with a limited number of antibodies, with the primary goal to facilitate gating, rather than as an unbiased way of gene expression analysis based on protein expression. In particular, we used CD45 to identify immune cells; CD19, CD20, CD14, and CD8 to exclude B cells, myeloid cells, and CD8+ T cells; CD3 and CD4 to identify CD4+ T cells; and CD45RA and CD45RO to distinguish between naïve and memory CD4+ T cells. This information was shown in Fig. S4 of the original manuscript (now Fig S6 of the revised manuscript). In addition, we included CD197 (CCR7), CD27, CD62L, CD127, CD279 (PD1), and CD49d as additional phenotyping markers and markers to help identify some cellular subsets like central memory T cells, effector memory T cells, and regulatory T cells. We have now clarified the purpose of these phenotyping markers in the text (Lines 564-568 in revised manuscript).

6. How was the cutoff for HIV low and high determined? Is this arbitrary, perhaps why a difference is not observed when stratifying by this category (Page 8)? This also drives home the fact we need more HIV RNA+ cells, not just increased reads from the same cells as shown in Fig 2C-D.

The cutoff for HIV low and high was based on the observed distribution of HIV RNA+ cells in our data, since we identified what seemed like a bimodal distribution of two populations of high vs low cells in the 2D dot plots shown in Fig. S3A and B of the original manuscript. The threshold of 50 reads was chosen and applied across all samples, as this seemed to be the threshold that distinguished the HIV RNA+ cells with lower vs. higher levels of HIV transcripts. Because a histogram depiction of the data may better illustrate the separation between the HIV_{low} vs. HIV_{high} populations, we have now added that visualization as a new plot in Fig. S5B. That visualization is copied below, where it is apparent, particularly when looking at the total population of HIV RNA+ cells, that there is a bimodal distribution of HIV RNA+ cells:

With regards to needing more HIV RNA+ cells and not just increased reads from the same cells, we note that HIV-seq does in fact increase the numbers of recovered HIV RNA+ cells as compared to standard 5' 10X sequencing. This result was demonstrated in Figure 2B in our original manuscript. We also note, as already discussed above, that our study reports the highest number of HIV RNA+ cells analyzed to date.

7. There seems to be a marked bias shown in Fig 2D indicating a 3' bias. Can the authors improve this by including probes for the 3' LTR region of the virus? Also given that there are many subtypes of HIV, can the authors expand this method to other non-B subtypes to make this approach more generalizable to other global subtypes?

We agree that additional probes targeting the 3' LTR region would be of interest to test for improvement, as would expanding the probes to enable analysis of other non-B subtypes and testing specimens obtained from non-B regions of the world. However, these aspects are well beyond the scope of the current study.

8. The authors have validated previous findings shown during viremia that TEMs have the most number of HIV RNA+ cells and express cytolytic markers in a pool of >1000 HIV RNA+ cells. However, their key findings at the ART suppressed timepoint is much weaker as only 26 cells were detected with HIV RNA. Additional samples to increase cell numbers at the ART timepoint like the viremic will be essential to have meaningful results for comparisons and title is thus overstated.

We would like to point out that we detected the highest number of HIV RNA+ cells reported to date in the context of ART suppression, and to date no published studies have compared HIV RNA+ cells in the context of ART suppression and viremia. One is limited by the low frequencies of HIV RNA+ cells that are present during ART suppression, and even if one were to spend years

adding a lot more donors to the analysis (which is well beyond the scope of this study), the numbers of HIV RNA+ cells would still overall be low due to the nature of the HIV reservoir.

Minor points:

1. Further, as seen in the supplementary tables, a lot of host transcripts are long-noncoding RNA and thus not polyadenylated are sequenced when using the default 10x platform and so authors should tone down their suggestions that only poly-A sequences are targeted.

While the 10X platform targets polyadenylated transcripts, it is known that the system is not completely clean and can capture some non-polyadenylated transcripts, likely through nonspecific priming or self-priming. We have now added a sentence stating this observation (Lines 106-107).

2. Page 7 lines 129-130: figure 2A simply shows cluster distributions on UMAs. This evidence is insufficient to support the assertion of “no difference in global gene expression.”

We agree with the reviewer that a more quantitative assessment of whether any transcriptional changes is elicited by HIV-seq is warranted. Hence, we performed differential gene expression (DEG) analysis on two patient specimens that were analyzed in the absence vs. presence of HIV-seq. These new results are now presented as volcano plots (Fig. S2 in the revised manuscript) and in new supplementary tables (Tables S3 and S4), and are depicted below:

In total, implementation of HIV-seq resulted in only 4 DEGs in PID1052 and 6 in PID8027. The only gene common to both, *AL138963.4*, is a long non-coding RNA³ with limited characterization. Given that such low numbers of DEGs are in line with what would be expected with technical replicates of scRNA-seq analysis of the same sample, we believe our original conclusion of there being no significant global gene expression difference between the two experimental conditions holds. These additional data are now discussed in the revised manuscript (Lines 129-133).

3. Pages 8,9 lines 174-182: can you show where these subclusters fall on your UMAP? Would go a long way to helping contextualize the data shown in subfigure 3B.

We have now added a visualization of the classical CD4+ T-cell subsets on the UMAP in Fig. S7A.

This visualization demonstrates that the majority of HIV RNA+ cells overlap with the Tem population, supporting our statement that HIV+ cells are enriched in Tem cells.

4. Figure 3F: pie charts are not ideal way to show data. It is very difficult to interpret changes in relative representation for clusters other than 1 and 2.

We have now replaced the pie charts with stacked bar charts to indicate percentage distribution, similar to panel C, as a clearer representation of the data.

5. Are there sufficient cells to do TCR analysis?

Unfortunately, VDJ libraries were not generated for this study, so TCR analysis could not be performed. As mentioned in the discussion of our initial manuscript (Line 468-471, Lines 501-504 in revision), VDJ sequencing is compatible with the HIV-seq platform and should be integrated in future studies.

6. Were HIV transcripts included while clustering?

HIV transcripts were included in the clustering. The only genes removed were TCR genes, as they caused donor-specific clustering driven by TCR expression. As detailed in Response #11 to Reviewer #1, inclusion of HIV transcripts in the clustering did not alter cluster assignment nor distribution.

Reviewer #3

(Remarks to the Author):

References:

1. Bartz, S. R. & Vodicka, M. A. Production of high-titer human immunodeficiency virus type 1 pseudotyped with vesicular stomatitis virus glycoprotein. *Methods* **12**, 337–342 (1997).
2. Telwatte, S. *et al.* Heterogeneity in HIV and cellular transcription profiles in cell line models of latent and productive infection: implications for HIV latency. *Retrovirology* **16**, 32 (2019).

3. Touil, H. *et al.* A structured evaluation of cryopreservation in generating single-cell transcriptomes from cerebrospinal fluid. *Cell Rep Methods* **3**, 100533 (2023).
4. Chomont, N. *et al.* HIV reservoir size and persistence are driven by T cell survival and homeostatic proliferation. *Nat Med* **15**, 893–900 (2009).
5. Dai, X.-P. *et al.* Increased Platelet-CD4+ T Cell Aggregates Are Correlated With HIV-1 Permissiveness and CD4+ T Cell Loss. *Front. Immunol.* **12**, (2021).
6. Kiselev, V. Y., Andrews, T. S. & Hemberg, M. Challenges in unsupervised clustering of single-cell RNA-seq data. *Nat Rev Genet* **20**, 273–282 (2019).
7. George, A. F. *et al.* Deep Phenotypic Analysis of Blood and Lymphoid T and NK Cells From HIV+ Controllers and ART-Suppressed Individuals. *Front Immunol* **13**, 803417 (2022).
8. Yin, K. *et al.* Long COVID manifests with T cell dysregulation, inflammation and an uncoordinated adaptive immune response to SARS-CoV-2. *Nat Immunol* **25**, 218–225 (2024).
9. Chari, T. & Pachter, L. The specious art of single-cell genomics. *PLOS Computational Biology* **19**, e1011288 (2023).
10. Tran, H. T. N. *et al.* A benchmark of batch-effect correction methods for single-cell RNA sequencing data. *Genome Biology* **21**, 12 (2020).
11. Jaskowiak, P. A., Costa, I. G. & Campello, R. J. G. B. The area under the ROC curve as a measure of clustering quality. *Data Min Knowl Disc* **36**, 1219–1245 (2022).
12. McGinnis, C. S., Murrow, L. M. & Gartner, Z. J. DoubletFinder: Doublet Detection in Single-Cell RNA Sequencing Data Using Artificial Nearest Neighbors. *Cell Systems* **8**, 329-337.e4 (2019).
13. Chen, Y., Chen, L., Lun, A. T. L., Baldoni, P. L. & Smyth, G. K. edgeR v4: powerful differential analysis of sequencing data with expanded functionality and improved support for small counts and larger datasets. *Nucleic Acids Research* **53**, gkaf018 (2025).

14. Robinson, M. D., McCarthy, D. J. & Smyth, G. K. edgeR: a Bioconductor package for differential expression analysis of digital gene expression data. *Bioinformatics* **26**, 139–140 (2010).
15. Collora, J. A. *et al.* Single-cell multiomics reveals persistence of HIV-1 in expanded cytotoxic T cell clones. *Immunity* **55**, 1013-1031.e7 (2022).
16. Wei, Y. *et al.* Single-cell epigenetic, transcriptional, and protein profiling of latent and active HIV-1 reservoir revealed that IKZF3 promotes HIV-1 persistence. *Immunity* **56**, 2584-2601.e7 (2023).
17. Geretz, A. *et al.* Single-cell transcriptomics identifies prothymosin α restriction of HIV-1 in vivo. *Sci Transl Med* **15**, eadg0873 (2023).
18. Peripheral Blood | Whole blood | Handbook | Miltenyi Biotec | USA.
<https://www.miltenyibiotec.com/US-en/support/macs-handbook/human-cells-and-organs/human-cell-sources/blood-human.html>.
19. Su, L. F., Kidd, B. A., Han, A., Kotzin, J. J. & Davis, M. M. Virus-specific CD4⁺ memory phenotype T cells are abundant in unexposed adults. *Immunity* **38**, 373–383 (2013).

Top 20 genes clusters - Table showing the top 20 genes defining each cluster, identified using the FindAllMarkers function. These genes represent the most differentially expressed features distinguishing each cluster in our analysis.

myAUC	avg_diff	power	avg_log2FC	pct.1	pct.2	cluster	gene
0.193	-2.3348246	0.614	-5.0708663	0.141	0.683	0	NKG7
0.201	-1.7716477	0.598	-3.4886879	0.22	0.719	0	CCL5
0.791	0.43818525	0.582	0.64997967	1	0.999	0	RPL32
0.79	0.73823885	0.58	1.19528325	0.993	0.654	0	LTB
0.788	0.45908093	0.576	0.67657052	1	0.999	0	RPS12
0.777	0.34892301	0.554	0.51643183	1	1	0	RPL28
0.225	-1.6189556	0.55	-4.7931754	0.037	0.571	0	GZMH
0.773	0.39486338	0.546	0.59193811	1	1	0	RPL11
0.771	0.41437924	0.542	0.61109316	1	1	0	RPLP1
0.771	0.43842553	0.542	0.65612048	1	0.999	0	RPS8
0.77	0.38614109	0.54	0.57321055	1	0.999	0	RPL30
0.769	0.41228878	0.538	0.60948678	1	1	0	RPL13
0.769	0.38016739	0.538	0.57853135	1	0.999	0	RPS25
0.765	0.37794708	0.53	0.56470468	1	0.999	0	RPS28
0.765	0.43650916	0.53	0.6634181	1	0.999	0	RPS13
0.235	-2.2757943	0.53	-5.1860133	0.105	0.59	0	GNLY
0.763	0.39254818	0.526	0.58523634	1	0.998	0	RPL39
0.761	0.36442845	0.522	0.55275665	1	0.999	0	RPL12
0.239	-1.2019387	0.522	-3.2953726	0.146	0.606	0	CST7
0.76	0.37238775	0.52	0.55794255	1	0.999	0	RPS23
0.961	2.04460403	0.922	3.3499365	0.992	0.333	1	NKG7
0.942	1.67720456	0.884	2.78113042	0.992	0.401	1	CCL5
0.931	1.38402699	0.862	2.82552553	0.968	0.276	1	CST7
0.93	1.6042996	0.86	3.06857948	0.955	0.2	1	GZMH
0.091	-1.7298964	0.818	-3.3615096	0.318	0.921	1	LTB
0.9	1.45793818	0.8	3.17481355	0.889	0.123	1	FGFBP2
0.888	2.050575	0.776	3.41806489	0.89	0.269	1	GNLY
0.882	1.4440429	0.764	3.23313722	0.852	0.119	1	GZMB
0.879	1.00010397	0.758	2.15672145	0.934	0.368	1	EFHD2
0.873	1.1727343	0.746	2.61037603	0.879	0.179	1	PRF1
0.87	1.06539276	0.74	2.22083248	0.918	0.255	1	GZMA
0.862	0.50598516	0.724	0.74835661	1	1	1	HLA-B
0.859	0.59219136	0.718	0.88894118	1	1	1	HLA-C
0.854	0.52402468	0.708	0.76652955	1	1	1	B2M
0.844	0.72108267	0.688	1.14212626	1	0.97	1	SH3BGRL3
0.826	1.03353964	0.652	2.37453436	0.83	0.303	1	CTSW

0.821	0.71981885	0.642	1.32381237	0.97	0.79	1	HCST
0.821	0.59369248	0.642	0.92497755	1	0.992	1	PFN1
0.817	0.4426398	0.634	0.67058973	1	1	1	HLA-A
0.188	-0.9265131	0.624	-2.6190354	0.182	0.741	1	TCF7
0.079	-0.9846992	0.842	-1.4755021	0.999	1	2	RPL10
0.083	-1.006314	0.834	-1.4962126	1	1	2	EEF1A1
0.903	0.9402016	0.806	1.36245764	1	1	2	MALAT1
0.101	-0.8325547	0.798	-1.274441	0.999	1	2	RPS3
0.102	-0.8777045	0.796	-1.3238554	0.999	1	2	RPL30
0.108	-0.8416553	0.784	-1.2882958	0.997	0.999	2	RPS14
0.109	-0.9046846	0.782	-1.4185212	0.997	1	2	RPS13
0.109	-0.8919467	0.782	-1.3349694	0.999	1	2	RPS27
0.111	-0.7586369	0.778	-1.1759852	0.996	0.999	2	RPL26
0.113	-0.8297812	0.774	-1.2627057	0.998	1	2	RPS28
0.117	-0.9256606	0.766	-1.3816689	0.998	1	2	RPS12
0.118	-0.8496518	0.764	-1.3345021	0.997	0.999	2	RPS4X
0.118	-0.716584	0.764	-1.1271185	0.997	1	2	RPS7
0.124	-0.7352427	0.752	-1.1170065	0.999	1	2	RPL19
0.126	-0.8466805	0.748	-1.2632566	1	1	2	RPLP1
0.872	0.79121813	0.744	1.17196003	1	0.999	2	MT-CO2
0.13	-0.7557711	0.74	-1.1603904	0.998	1	2	RPL18A
0.131	-0.7614308	0.738	-1.1637982	0.998	1	2	RPS27A
0.134	-0.7521028	0.732	-1.1477035	0.999	1	2	RPL11
0.138	-0.7240252	0.724	-1.1379	0.994	0.999	2	RPL35A
0.98	2.29985632	0.96	5.50034457	0.968	0.031	3	TRBV20-1
1	2.82415375	1	6.94136211	1	0.017	4	TRBV7-2
0.999	2.41445434	0.998	6.38179835	1	0.011	5	TRBV5-1
1	2.60033664	1	6.71921488	1	0.011	6	TRBV7-9
1	2.38726432	1	6.42766259	1	0.009	7	TRBV3-1
1	2.49412821	1	6.70608967	1	0.004	8	TRBV10-3
1	2.58392136	1	6.82737705	1	0.005	9	TRBV12-3
1	2.4313496	1	6.55747636	1	0.005	10	TRBV4-2
0.851	1.33492206	0.702	2.09668939	0.995	0.979	11	GAPDH
0.849	0.87494011	0.698	1.33647809	1	0.994	11	PFN1
0.849	0.94916228	0.698	1.40130609	1	0.999	11	ACTB
0.848	0.82266214	0.696	1.37226808	0.99	0.934	11	CORO1A
0.153	-1.0057309	0.694	-1.4652689	0.988	1	11	MALAT1
0.838	1.13179836	0.676	2.21725103	0.899	0.529	11	COTL1
0.826	0.76453967	0.652	1.21133529	0.999	0.987	11	CFL1
0.176	-0.8627951	0.648	-1.5187685	0.519	0.977	11	TXNIP
0.823	1.0467313	0.646	1.65300311	0.989	0.964	11	ACTG1

0.822	0.76551645	0.644	1.40478496	0.958	0.795	11	ARPC1B
0.181	-0.6645367	0.638	-1.0099831	0.986	0.999	11	RPL34
0.819	0.70122999	0.638	1.18757261	0.989	0.94	11	MYL6
0.817	1.56780111	0.634	3.1211505	0.839	0.472	11	HMGB2
0.816	0.96359613	0.632	2.20255219	0.836	0.437	11	H2AFV
0.199	-0.798039	0.602	-1.4934095	0.49	0.94	11	TSC22D3
0.8	0.96999467	0.6	1.67887492	0.969	0.891	11	HMGB1
0.2	-0.675725	0.6	-1.2437166	0.602	0.982	11	PNRC1
0.798	0.79754462	0.596	1.69465526	0.869	0.58	11	COX8A
0.797	0.6773138	0.594	1.41343343	0.901	0.611	11	PPP1CA
0.796	0.77154122	0.592	1.83685837	0.825	0.451	11	COX5A
0.833	1.98715108	0.666	5.58035609	0.677	0.025	12	NRGN
0.795	1.74946172	0.59	5.29218694	0.602	0.021	12	PPBP
0.776	1.10433395	0.552	4.22500165	0.56	0.013	12	CAVIN2
0.751	1.19282288	0.502	4.43816744	0.509	0.01	12	TUBB1
0.747	1.01356534	0.494	4.01214474	0.504	0.015	12	PF4
0.726	0.67118	0.452	3.34235889	0.456	0.005	12	GP1BB
0.718	1.22921748	0.436	3.6513003	0.516	0.111	12	HIST1H2AC
0.717	0.60391533	0.434	3.14175127	0.439	0.006	12	SPARC
0.713	0.70478873	0.426	3.36384472	0.433	0.01	12	PRKAR2B
0.707	0.69500493	0.414	3.20714262	0.431	0.021	12	CLU
0.707	0.65652881	0.414	1.1512443	0.94	0.873	12	TAGLN2
0.982	2.58148531	0.964	6.74933735	0.965	0.007	13	TRBV7-3
0.978	2.86109906	0.956	4.44292608	0.999	0.859	14	CD74
0.964	2.71692735	0.928	5.07854232	0.975	0.244	14	HLA-DRB1
0.964	3.20280084	0.928	6.51754877	0.95	0.102	14	HLA-DRA
0.916	1.75660682	0.832	5.55305819	0.832	0.004	14	SPI1
0.91	3.51891255	0.82	7.85394355	0.829	0.038	14	CST3
0.903	1.75353154	0.806	3.99502665	0.864	0.22	14	TYMP
0.901	3.27208046	0.802	7.77844856	0.805	0.016	14	LYZ
0.899	2.80734641	0.798	6.85398105	0.808	0.037	14	IFI30
0.899	1.46393418	0.798	2.75337003	0.923	0.531	14	COTL1
0.899	2.46111843	0.798	4.65103957	0.912	0.334	14	HLA-DPA1
0.894	1.67504181	0.788	3.3556875	0.9	0.461	14	PSAP
0.894	1.90737662	0.788	4.89750714	0.813	0.088	14	AIF1
0.892	1.25993186	0.784	1.92492655	0.997	0.997	14	FTH1
0.888	1.54412487	0.776	4.27656703	0.813	0.113	14	GRN
0.882	1.85444593	0.764	5.17524027	0.776	0.024	14	FCER1G
0.877	1.04170704	0.754	1.95800423	0.958	0.765	14	GABARAP
0.876	2.27183152	0.752	5.00069411	0.777	0.061	14	TYROBP
0.875	1.59050928	0.75	3.61561266	0.853	0.295	14	CTSS

0.873	1.55471984	0.746	4.57923166	0.77	0.067	14	LST1
0.872	2.04287751	0.744	3.98678385	0.884	0.352	14	HLA-DPB1

Reviewer #1 (Remarks to the Author):

It is a pity that instead of addressing concerns and improving rigor, the authors did not respond sufficiently to reviewer suggestions. This manuscript, despite presenting an exciting and innovative spike-in strategy, unfortunately did not demonstrate the rigor compared with other manuscripts at Nature Communications.

We had applied rigorous methods to analyze the scRNA-seq data, and had performed new experiments to demonstrate lack of detection of HIV RNA upon HIV-seq analysis of cells from people without HIV. As detailed further below, we have ensured that high rigor was applied in all aspects of the study, including adding a third uninfected donor and including details on all the numbers of cells sequenced from HIV- and HIV+ samples, which were similar (Lines 126-128, 139-142, 147-152, 170-172, 263-265, 1029-1031). Although the number of clusters is not an issue of rigor, we have also increased the resolution of the clusters, as suggested by the reviewer (lines 235-249, 275-293; Fig. 2F, S9, S10, 3E). We are not aware of any additional issues with rigor.

Point 1. Sorting pure populations of HIV+ samples is technically feasible (PMID: 30282021). The authors did not attempt to address this point.

There appears to have been a mis-understanding, as we were not making the case that one cannot use a FACS machine to sort HIV+ cells (we agree it can be done, we ourselves have extensively sorted HIV-infected primary cells, e.g., PMIDs 39798087, 28746881). Rather, we were making the case that the “100% HIV infection” that the reviewer stated is not feasible to achieve with primary cells, since the initial infection frequencies are generally low (<10%) and the sort purity maxes out at around 95%, so there will be uninfected cells in the mix. As one cannot know the exact numbers of true infected cells in these models, it will be difficult to establish the “upper end” of what corresponds to detection of 100% of the infected cells. It was because of this that we had discussed the notion of using clonally-derived cell line models (e.g., JLATs) to establish what would be a 100% detection rate.

But more importantly, and which we apologize for not having clarified in our original response, there is a critical problem with trying to demonstrate that HIV-seq improves detection of HIV RNA by performing a head-to-head comparison of conventional 10X vs. HIV-seq on either cell lines or even sorted HIV-infected primary cells. The problem is that the types of HIV transcripts from cell line or primary cell models of HIV infection do not recapitulate the types of HIV transcripts from ART-suppressed people with HIV (PWH) – in particular, these *in vitro* models do not replicate the excess of non-polyadenylated HIV transcripts that are observed *in vivo*. We have previously compared the levels and patterns of different HIV transcripts in unstimulated cells from ART-suppressed PWH to those measured in *in vitro* models of HIV persistence, including cell lines (7 different latently HIV-infected cell lines and the productively HIV-infected 8E5 cell line¹) as well as four different primary cell models of latent HIV infection². The blocks to HIV transcription were different in each cell line or primary cell model, but

none of them showed the vast excess of non-polyadenylated HIV transcripts (block to HIV transcriptional completion) that we observed in cells from PWH. Specifically, in cells from ART-suppressed PWH, the ratio of 5' elongated to polyadenylated (completed) HIV transcripts is about 10%, whereas in all cell lines and primary cell models that we have studied, this ratio is at least 60% and usually close to 100%^{1,2}. In these cell lines and primary cell models, most HIV transcripts that have extended past the 5' LTR have a polyA tail, and therefore would be expected to be captured by the conventional 10x approach using a poly-dT. Hence, one would not necessarily expect any advantage to using capture primers located between the 5' LTR and the polyA tail, as used in HIV-seq. By contrast, HIV-infected cells in PWH – including in the context of ART suppression as well as during active viremia^{3,4} – include high proportions of incomplete transcripts, consistent with our ability to improve detection of HIV transcripts when spiking in the HIV-seq primers targeting non-polyadenylation signals. In summary, our HIV-seq was specifically designed to increase the capture of non-polyadenylated HIV transcripts, which are primarily observed *in vivo* but not *in vitro*, and the best test of our method is to use actual samples from PWH, as performed in this manuscript. This is the most rigorous approach and the gold standard, and the only way to really test the approach. We apologize for not having been clearer on this point in our original rebuttal, when we had simply stated that the *in vitro* models do not recapitulate *in vivo* infected cells – we were referring specifically to the types of HIV transcripts that are present, which we hope we have now clarified.

Point 3. While this reviewer appreciates the use of uninfected cells as a control as requested, what's the number of cell sequenced, and how is the number comparable with the HIV+ samples? It is showing a lack of rigor.

We apologize for leaving out these details. Because the numbers of cells sequenced is standard and matched the numbers sequenced for the HIV+ samples, we had left out those details. But we agree that for completeness sake and full transparency, we should share all these details. For the two uninfected donors we had analyzed by HIV-seq, we had sequenced 7,260 and 5,998 CD4+ T cells, which is within the range of cell numbers sequenced from the HIV+ study participants (median of 7,011 cells sequenced per person; range 4,566-14,556). Because the numbers of cells sequenced for one of the uninfected donors was slightly on the lower end relative to the HIV+ samples, we have now sequenced cells from an additional (third) uninfected donor, from whom we analyzed 8,753 cells. In the three uninfected donors we analyzed by HIV-seq (where we sequenced 7,260, 5,998, and 8,753 cells), we did not identify a single HIV RNA read, whereas in all of the samples from the HIV+ donors (where similar numbers of cells were sequenced for each donor) we identified HIV RNA+ cells. These data clearly demonstrate that HIV-seq does not result in false positive reads. These cell numbers are now detailed in lines 139-42 and the revised figure legend of Figure S4, while cell numbers from the HIV+ participants are listed in lines 126-128, 147-152, 170-172, 263-265, and 1029-1031.

Point 7. For CD4 T cell resolution: the authors stated to stay with current resolution of 6 clusters without minimal biological annotations. In other existing publications, there are

10 clusters in PMID 35320704, 14 clusters in PMID 36070690, and 15 clusters in PMID 37922905. While it is not the number of cluster that matters - it is whether the authors examined and dissected the biology. See PMID: 33879890, Nature Med - 10 clusters, including naive, central memory, effector memory, Th1, Th2, Th17, Treg, proliferating, in the peripheral blood. Leaving the annotation as cluster numbers without resolution to these essential peripheral blood CD4 cell types does not bring insight to the field. Please at least follow cell type annotations by PMID: 33879890.

We appreciate and thank the reviewer for bringing up other studies that had used biological knowledge to justify increasing the numbers of clusters. Indeed, in the field there is currently lack of consensus as to how best to define cluster resolution. While we still think it is important to implement objective, statistics-based methods to define cluster resolution as suggested by others⁵, and as presented recently at the bioinformatics-based Systems for Molecular Biology (ISMB) conference⁶, and as we recently implemented in other high-parameter single-cell sequencing studies^{7,8}, we agree that bringing in biological knowledge is also important to establish biologically meaningful clusters.

To bring both approaches together, we first implemented our statistics-based clustering method, and then selected the two largest clusters (clusters 1 and 2) that drove the underclustering for further subclustering using the same optimized resolution applied in the statistics-based approach. We then considered each of the subclusters as clusters, and added them to the remaining clusters that we deemed were not under-clustered (clusters 3-6 in original annotation). This approach resulted in a total of 13 clusters. When manually annotated, we could break down the clusters into the following subsets (note cluster numbers have been re-assigned, as there are now 13 clusters):

- Cluster 1: Long-lived CD4+ T cells
- Cluster 2: Tcm
- Cluster 3: Non-cytotoxic Tem
- Cluster 4: ISG+ long-lived CD4+ T cells
- Cluster 5: *GZMK*+ Ttm
- Cluster 6: Cytotoxic Tem
- Cluster 7: *KLRD1*+ cytotoxic CD4+ T cells
- Cluster 8: Tem
- Cluster 9: Long-lived cytotoxic CD4+ T cells
- Cluster 10: ISG+ Proliferating CD4+ T cells
- Cluster 11: Platelet-CD4+ T cell aggregates
- Cluster 12: Other CD4+ T cells
- Cluster 13: Activated CD4+ T cells

We have incorporated the new clustering and cluster-distribution analyses into the Results (lines 235-249, 275-293), main figures (revised Fig. 2F and Fig. 3E), and supplementary figures (Fig. S9, S10) as well as the new Supplementary Table 6, and we revised the Methods and figure legends accordingly. Importantly, the main overall conclusions of our study remain unchanged: during viremia, HIV RNA+ cells

predominantly reside in CD4+ T cells with a cytotoxic (high granzymes A/B/H/M, perforin, granulysin) and Th1 signature, whereas during ART suppression., they are found not in cytotoxic CD4+ T cells but rather predominantly in a long-lived population of memory CD4+ T cells. With the updated clustering, however, we were able to gain additional insights into the features of HIV RNA+ cells during ART-suppression, in that we found enrichment of HIV RNA+ cells in a second cluster that, in addition to having features of stemness, displays high expression of *MALAT1*, a long non-coding RNA previously shown to promote HIV transcription and infection⁹. These new findings have been added to the Results section of the revised manuscript (Lines 285-293).

Point 12: We requested removal of Figure 1B, which is a commercial 10x Genomics illustration - but Figure 1B remains in this version. The current Figure 1 remains extremely weak. I have never seen any publication with such a weak Figure 1 - probe design, that's it, no data, as a major Figure.

In our first revision, we removed the portion of Figure 1B that illustrated the commercially available 10X platform pipeline while retaining the workflow unique to HIV-seq. We note that our revised Figure 1B from the previous submission was not a 10X Genomics illustration; it had showed an image of a 10X controller and an illustration completely created by us (not derived from any 10X Genomics literature) showing the relationship of our HIV capture sequences to the PCR handle and to their binding transcripts. In response to the reviewer's request that this not be a main figure, however, we have now moved this schematic we created to a new supplementary figure (Fig. S2). In addition, we have merged the original Figure 1A with the previous Figure 2. These panels now appear together as the new, consolidated Figure 1 in the revised manuscript.

Reviewer #2 (Remarks to the Author):

The authors now provide the frequency of HIV RNA+ cells and the total number of CD4 T cells assessed for each donor and experiment. From these numbers a z-test performed by this reviewer, found no significant differences ($P > 0.05$) for HIV RNA proportions with and without HIV-SEQ for all comparisons. It will be important to justify more precisely than is currently provided, on what basis the inclusion of HIV-1 probes is claimed to increase detection of HIV RNA+ cells.

Given the extremely high cost of scRNA-seq, the head-to-head comparison of HIV-seq vs. conventional 10x was performed on cells from a limited number of study participants. The z-test mentioned by the reviewer should not be applied to our data, since as it assumes that the data follows a normal distribution and that the population variance is known (not calculated from the data) or that the sample size is large ($n > 30$) - conditions that are not met in our study. While the number of study participants does not support formal statistical comparison of HIV RNA+ cell frequencies, we found that the frequency of HIV RNA+ cells increased by 26% and 44.7% in the two donors when using HIV-seq (Lines 147-152), which supports the idea HIV-seq improves detection of HIV-infected cells. To further demonstrate that the HIV capture primers helped to

increase the detection of HIV RNA, we used the Mann-Whitney test to compare the number of HIV reads per cell detected with HIV-seq compared to the conventional 10x. We found that HIV-seq doubled the mean number of reads per cell, and the difference was very statistically significant ($P < 0.01$; Fig. 1D in latest version of manuscript), demonstrating with scientific rigor that HIV-seq enhances detection of HIV transcripts. We now acknowledge that the small number of study participants did not support formal statistical comparison of HIV RNA+ cell frequencies (lines 152-153), and we have revised the text of the manuscript to emphasize that it is the number of reads per cell that is statistically significant (lines 94, 152-154, 162-164, 389-390, 395).

Consequently, although this paper is the second scRNA-seq study with >1000 HIV RNA+ cells detected, the authors have not currently demonstrated that this is because of their method.

To our knowledge, the only published studies specifically analyzing HIV RNA+ cells from people with HIV by scRNA-seq during viremia are the three we cited in the Discussion (Collora et al. [25], Geretz et al. [32], and Wei et al. [51]). These studies reported a maximum of 256 HIV RNA+ cells (81 cells, 164 cells [with one highly viremic donor further sequenced to gain an additional 223 HIV RNA+ cells] and 256 cells, respectively).

It is possible that the >1,000 HIV RNA+ the reviewer referred to is from Figure 4 of the Geretz et al (reference 32) study, which was pointed out to the editor. In that figure, there were numbers above the bar graphs corresponding to individual participants, one of which had a number greater than 1,000 under the category of “vRNA+”. While we could not find definitions for what the numbers above the bars refer to within the Results or Figure Legends of the paper, given that this figure was brought up, we presume they refer to numbers of HIV RNA+ cells that were identified. Going with this assumption, we have now acknowledged in our revised Discussion that one prior study has detected >1,000 HIV RNA+ cells during viremia (Lines 401-403). We would like to point out that we never made the claim that it is only because of HIV-seq that one can analyze >1,000 HIV RNA+ cells. It is simply a method we present of interest for the field that increases the efficiency of HIV RNA detection, and which we applied on cells from PWH and whose datasets we then further analyzed. We have now added a sentence in lines 406-407 acknowledging that the high number of HIV RNA+ cells identified may reflect differences in the underlying frequency of HIV RNA+ cells as well as differences in HIV RNA capture.

Detecting further HIV RNA+ cells at the viremic timepoint has only added incremental insight into HIV RNA+ cells.

With due respect, and as detailed in our manuscript, we do not believe our analysis of the viremic timepoint only added incremental insight into HIV RNA+ cells. New insights relating to HIV RNA+ cells during viremia included:

- 1) reduced expression of *S100A8*, *S100A9*, *APOBEC3A*, and *SERPINA1* in HIV RNA+ cells relative to HIV RNA- cells (lines 212-219, 437-452)
- 2) upregulation of *CXCR6* expression and an enhanced chemokine signaling pathway in HIV RNA+ cells compared to HIV RNA- cells (lines 224-225, 229-231)
- 3) upregulation of both NFAT and PKC pathways in HIV RNA+ cells compared to HIV RNA- cells (lines 226-229)

Our data also confirmed prior findings on HIV RNA+ during viremia (also important for the field are confirmatory studies in different cohorts), in particular the demonstration that:

- 1) HIV RNA+ cells exhibit a heterogeneous phenotype but are enriched among Tem cells
- 2) HIV RNA+ cells are enriched among cells with a cytotoxic/Th1 phenotype

In addition, we report new insights into the mapping of HIV RNA reads detected in single cells before and after ART (lines 160-162), the phenotypes and transcriptomes of HIV RNA+ cells during ART suppression (lines 260-301), and how they differ from HIV RNA+ cells during viremia (lines 341-385, 472-476, 477-526). Together with the new method used to increase detection of HIV RNA during scRNA-seq, our manuscript has added much more than incremental knowledge. It should be noted that the description of our new method could comprise an entire manuscript by itself, yet we have combined it with a wealth of new data on the transcriptomes of cells in longitudinal samples from viremic and ART-suppressed people with HIV.

Not using the ADT data in their differential analysis and generating TCR libraries are missed opportunities.

As indicated in the prior rebuttal, we had used the ADT data primarily for cell phenotyping and gating rather than analysis of differential gene expression because the antibody panels were pre-selected and comprised a relatively small number of targets, as compared to the unbiased and full transcriptome data from scRNA-seq. However, we appreciate the reviewer's suggestion and agree that the limited protein expression data can also be informative for differential gene expression analysis. Some markers—CD8, CD19, CD20, and CD14—were excluded from this analysis, as they were not relevant for the CD4+ T cell population given that they were used to gate out CD8+ T cells, B cells, and monocytes. We performed differential protein expression analyses using the remaining 11 markers and have now included these results in the revised manuscript (Lines 200-207, 268-270, and 344-347; new Supplementary Figures S8, S11, and S14). We found that during viremia, HIV RNA+ cells express lower levels of CD45RA and higher levels of CD45RO compared to HIV RNA- cells across all donors, consistent with our original reports that HIV RNA+ cells are depleted among naïve CD4+ T cells (which express more CD45RA) and enriched in memory CD4+ T cells (which express more CD45RO) (new Fig. S8). We also found some proteins that were differentially expressed in HIV RNA+ cells as compared to HIV RNA- cells, but only in a subset of participants – these included CD279 (PD1; upregulated in HIV RNA+ cells in

2/4 people), CD27 (downregulated in HIV RNA+ cells in 3/4 people), and CD49d (upregulated in 3/4 people; new Fig. S8). By contrast, during ART suppression we did not find any ADT features consistently significantly differentially expressed among HIV RNA+ vs RNA- cells. However, comparing HIV RNA+ cells during viremia versus ART suppression revealed lower CD4 expression in the context of active viremia, most likely reflecting either increased production or enhanced functionality of Nef protein, which is known to downregulate cell-surface CD4 expression (new Fig. S14). These interesting new data have now been added to our manuscript (Lines 344-347, new Supplementary Fig. 14).

Regarding TCR library generation, we agree this is a valuable tool, as it can inform about clonal expansion dynamics and allow for prediction of antigen specificities. We had acknowledged in the Discussion that multiplexing with TCR library generation is an important next step (Lines 528-532 revised manuscript), but note that these studies are well beyond the current one, which was not focused on clonal expansion dynamics or antigen specificities.

Although the authors sought to perform ‘a meaningful analysis of HIV RNA+ cells from ART-suppressed PWH’ (line 99), they were only able to detect 25 HIV RNA+ cells at this timepoint in 3 donors which did not allow for a rigorous analysis.

Although 25 HIV RNA+ cells is not high, we note that this number is the highest analyzed to date from blood of ART-suppressed people with HIV. Throughout the manuscript, we had been careful not to overstate our findings. We explicitly indicated when analyses are exploratory and noted that such results should be interpreted with caution. We also made sure that the results presented were rigorous and made use of appropriate statistical tests and data visualization approaches.

Finally, and most importantly, reluctance to remove HIV while clustering,

The reviewer seems to be referring to a point brought up by the other reviewer (Reviewer #1, Point #11 from the prior revision). As detailed in our previous response (Response #11 to Reviewer #1), we had performed an analysis showing that removing HIV transcripts had minimal impact on clustering, given that 96% of the cells retained their original cluster identity. We had stated that this was our justification for not removing HIV transcripts while clustering, and in our original response had offered to include these analyses if Reviewer #1 considered it important. Reviewer #1 made no further comments on the matter, so we assume Reviewer #1 does not feel it to be important.

However, given Reviewer #2’s new concern on Reviewer #1’s prior comment, we have replaced our initial clustering (performed with HIV genes included) with the version generated after excluding HIV genes from the clustering step (new Fig. 2F and Fig. 3E). The Methods section has been updated accordingly (Lines 658) to reflect this change. As expected, no conclusions are altered from this change, since 96% of the cells retained their original cluster identity.

ineffective batch correction,

This is another comment that was from the other reviewer (Reviewer #1, Point #10 from the prior revision). Reviewer #1 had requested: “Please use a different batch effect correction method (such as Harmony) and replot Figure S6 to see if there are still batch effects.” We addressed this concern in our initial response (Response #10 to Reviewer #1) by showing that, even after applying the Harmony batch correction method as suggested, donor-to-donor differences persisted. These results were presented as Figure S6 in our prior revision (copied again below for easy reference).

The differences observed with Harmony were similar to those seen with Seurat’s batch correction. As explained in our response to Reviewer #1, given that inter-donor variability is expected in primary human samples, we believe these differences reflect true biology rather than batch effects, and that no batch correction methods would be able to correct for these biological differences. Reviewer #1 made no further comments on the matter. However, given this new concern from Reviewer #2 on the original comment from Reviewer #1, we have replaced the plots in new Fig. S12 with the Harmony batch-corrected version.

and the absence of code to replicate their findings makes it challenging to evaluate the findings and suggests a lack of rigor.

We note that the code for the quality control steps had already been shared previously, and the other steps involving code—such as clustering, differential expression analysis, and cell distribution-related statistics—were described in the Methods section, as is the standard for scientific manuscripts.

In this second revision, we now show all code by providing a link to a Github repository (<https://github.com/gladstone-institutes/HIV-seq- Viremic vs. ART>) that has been made available to the public.

This repository contains all scripts used in the manuscript:

- Step0_QCsteps.v8.R – used for QC steps after CellRanger and for formatting the data for SeqGeq software
- Step1a.R and Step1b.R – used to format cell count data from SeqGeq into a Seurat object
- Step2.R – used to analyze cell proportions for Figures 2C, 3C, 4B, and 4D
- Steps3, 4, and 5 – used for normalization, visualization, and batch correction to prepare the data for visualization
- Step6.R and Step7.R – used for cluster resolution optimization and cluster definition in Figures 2F and 3E
- Step8.R – used for differential gene expression analysis in Figures 2D, 3D, 4C, and 4E

The Data Code and Availability Statement has been updated to include this link (lines 727-733).

Only some data generated in the study was submitted to GEO.

Our original GEO repository (GSE266455) only included the sequences generated from the first version of the manuscript. We apologize that we had forgotten to upload the new data generated for the previous round of revisions. To publicly share the new data generated for the revision, we have now created a new GEO repository (GSE305352). This repository contains the two new uninfected donors analyzed as described in the previous submission, in addition to one additional uninfected donor that we included (since one of the uninfected donors sequenced was on the lower end number of cells sequenced) as detailed in our response to Reviewer #1.

This sequence data can be accessed using the following links and tokens:

To review GEO accession GSE305352:

Go to <https://www.ncbi.nlm.nih.gov/geo/query/acc.cgi?acc=GSE305352>

Enter token ulwfmysesxxgtjgn into the box

To review GEO accession GSE266455:

Go to <https://www.ncbi.nlm.nih.gov/geo/query/acc.cgi?acc=GSE266455>

Enter token azodqesezbelnuj into the box

The Data Code and Availability Statement has been updated to include these links (lines 727-733).

Improving existing methods is essential, but dependent on sufficient transparency for others to validate results, which is not currently available here.

We agree that transparency is essential for validating scientific results. With the full set of codes now shared, the addition of the negative control data to the GEO repository, and our detailed responses to the reviewer's remaining questions, we believe that we have now provided sufficient transparency to enable others to fully assess and reproduce our findings.

Reviewer #3 (Remarks to the Author):

-
1. Telwatte, S. *et al.* Heterogeneity in HIV and cellular transcription profiles in cell line models of latent and productive infection: implications for HIV latency. *Retrovirology* **16**, 32 (2019).
 2. Moron-Lopez, S. *et al.* Human splice factors contribute to latent HIV infection in primary cell models and blood CD4+ T cells from ART-treated individuals. *PLoS Pathog* **16**, e1009060 (2020).
 3. Janssens, J. *et al.* Longitudinal changes in the transcriptionally active and intact HIV reservoir after starting ART during acute infection. *J Virol* **99**, e0143124 (2025).
 4. Janssens, J. *et al.* Differential decreases in various HIV DNA regions and HIV transcripts after ART initiation during chronic infection. *J Virol* **99**, e0068325 (2025).
 5. Kiselev, V. Y., Andrews, T. S. & Hemberg, M. Challenges in unsupervised clustering of single-cell RNA-seq data. *Nat Rev Genet* **20**, 273–282 (2019).
 6. Gill N & Thomas R. Optimizing Clustering Resolution for Multi-subject Single Cell Studies. Presented at Intelligent Systems for Molecular Biology (ISMB). <https://github.com/gladstone-institutes/clustOpt> (2025).
 7. George, A. F. *et al.* Deep Phenotypic Analysis of Blood and Lymphoid T and NK Cells From HIV+ Controllers and ART-Suppressed Individuals. *Front Immunol* **13**, 803417 (2022).
 8. Yin, K. *et al.* Long COVID manifests with T cell dysregulation, inflammation and an uncoordinated adaptive immune response to SARS-CoV-2. *Nat Immunol* **25**, 218–225 (2024).
 9. Wang, M.-R. *et al.* LncRNA MALAT1 Facilitates HIV-1 Replication by Upregulation of CHCHD2 and Downregulation of IFN-I Expression. *Mol Cell Proteomics* **24**, 100997 (2025).

POINT-BY-POINT RESPONSE TO REVIEWERS

Reviewer #1 (Remarks to the Author):

This reviewer recognizes the effort that the authors attempted to revise the manuscript, although the quality of the bioinformatic analysis is not up to the current standard of the field.

To our knowledge, every aspect of the bioinformatic analysis was performed according to the current gold standard of the field.

These are previously raised points that were not addressed.

Figure 3A and 4E

Please annotate the clusters, instead of 1, 2, 3, 4, 5, 6.

This point was already addressed. We believe that this reviewer was looking at an older version of the figures and not the most recent ones. The referenced number of clusters (6) and figure numbers (Fig. 3A and 4E) were from an older version of the manuscript on which the reviewer requested cluster annotation. In the most recent version of the manuscript, annotations were provided for clusters 1-13 in Fig. 2F and 3E, respectively .

Please integrate uninfected data into Figure 5.

Figure 5A: one UMAP of batch-effect-corrected single cell data including data from viremic, suppressed, and uninfected donors.

Figure 5B and 5D: Add uninfected donor data to the bar graph, as a reference to see how HIV affects these T cells.

Figure 5C: Add differential gene expression volcano plot of viremia vs uninfected, suppressed vs uninfected

Figure 5E: Add uninfected donor data to the bar graph, as a reference to see how HIV affects these T cells.

This comment from Reviewer #1 appears to be a completely new request, and appears to be referencing an older version of the manuscript. In the latest version of the manuscript, there is no Fig. 5 and main figures end at Fig. 4. While we could try to incorporate data from the uninfected donors, there is not enough space to include the data in this figure, and it does not make sense scientifically. The 3 uninfected donors were analyzed simply to demonstrate that the HIV-seq technique does not result in false positive detection of HIV transcripts. The figure referred to by the reviewer compares paired samples from the same people before vs. after ART suppression. Adding in the 3 uninfected donors will result in a very mixed set of comparisons that would be hard to interpret: we would be comparing longitudinal data from the same infected people at two different timepoints (during viremia vs. ART suppression) but also trying to compare data from those two timepoints cross-sectionally with data from 3

different HIV-uninfected donors. Perhaps even more importantly, the 3 uninfected donors were in no way matched to the HIV+ individuals analyzed in this study (in age, sex, and other clinical aspects). Our study had not set out to compare HIV+ people to HIV- people, which would require a high "n" and a completely different study design. This comparison should be reserved for a future study.